# Mexican Emotional Speech Database Based on Semantic, Frequency, Familiarity, Concreteness, and Cultural Shaping of Affective Prosody

**Mathilde Marie Duville *** , **Luz María Alonso-Valerdi** and **David I. Ibarra-Zarate**

Escuela de Ingeniería y Ciencias, Tecnologico de Monterrey, Ave. Eugenio Garza Sada 2501, Monterrey 64849, Mexico; lm.aloval@tec.mx (L.M.A.-V.); david.ibarra@tec.mx (D.I.I.-Z.)
\* Correspondence: a00829725@itesm.mx

**Abstract:** In this paper, the Mexican Emotional Speech Database (MESD) that contains single-word emotional utterances for anger, disgust, fear, happiness, neutral and sadness with adult (male and female) and child voices is described. To validate the emotional prosody of the uttered words, a cubic Support Vector Machines classifier was trained on the basis of prosodic, spectral and voice quality features for each case study: (1) male adult, (2) female adult and (3) child. In addition, cultural, semantic, and linguistic shaping of emotional expression was assessed by statistical analysis. This study was registered at BioMed Central and is part of the implementation of a published study protocol. Mean emotional classification accuracies yielded 93.3%, 89.4% and 83.3% for male, female and child utterances respectively. Statistical analysis emphasized the shaping of emotional prosodies by semantic and linguistic features. A cultural variation in emotional expression was highlighted by comparing the MESD with the INTERFACE for Castilian Spanish database. The MESD provides reliable content for linguistic emotional prosody shaped by the Mexican cultural environment. In order to facilitate further investigations, a corpus controlled for linguistic features and emotional semantics, as well as one containing words repeated across voices and emotions are provided. The MESD is made freely available.

**Keywords:** affective computing; audio database; cross-cultural; machine learning; Mexican Spanish; emotional speech; paralinguistic information; discrete emotions

## 1. Introduction

Human–computer affective interactions have been extensively studied, with many applications developed in a wide variety of fields. A case in point is speech emotion recognition, which aims to design artificial intelligence mathematical models able to predict human emotional states from affective voice signal processing [1]. Such systems are nowadays useful in security, healthcare, videogaming and mobile communications, where the objective and rapid assessment of emotional states enriches the user environment [2]. Given the fundamental role of emotion communication between conspecifics to establish social bounds, it is not surprising that non-lexical patterns including rhythm, frequency content, volume, breathiness and roughness are cross-culturally defined as universal cues established by human beings to express and recognize emotions [3]. Nevertheless, previous evidence pointed out an in-group advantage for individuals from the same ethnic group to recognize emotions from vocal utterances [4,5]. In this respect, the dialect theory argues that cultures are shaped by subtle differences in the ways to express and recognize emotions, which become greater the more the ethnic groups are culturally distant [6,7]. Those slight cross-cultural differences constitute a challenging issue for speech emotion recognition algorithms that are sensitive to even subtle acoustic features [8]. Therefore, the elaboration of databases adapted to specific cultures and languages is an urgent current need for the correct development of affective recognition [2,9].

Emotions are defined by neurophysiologists as deviations from the homeostasis of the central and peripherical nervous systems elicited by emotionally charged stimuli [10]. Those responses are the result of the processing of at least two emotional dimensions: valence and arousal. While valence refers to pleasantness (i.e., negativity versus positivity), the arousal dimension gives information about the level of activation (i.e., how calm or excited one feels when processing the stimulus) [11]. In situations of high arousal when experiencing anger, fear or excitement, the interplay between the increased activity of the sympathetic nervous system and the vagal activity leads to high vasoconstriction and increased sweat gland activity [12]. This increased sympathetic activity elicits changes in respiratory movements, high subglottal pressure and dryness of the mouth, leading to fast and loud speech, and high energy in higher frequencies [2,13]. On the other hand, in low arousal situations, such as experiencing sadness, blood pressure and heart rate decrease, and the activity of the salivary glands increases, producing a slow speech rate, higher energy in low frequency bands and medium-low intensity [2,14]. As a consequence, speech features extracted to characterize emotion prosody refer to frequency, intensity, energy and duration. However, extracting efficient features to accurately describe emotional information in the voice has been a challenging issue in past investigations [15]. As most classification algorithms are sensitive to overlapping information and redundancy, it is indispensable to efficiently select the acoustic parameters in order to achieve higher accuracy and avoid overfitting biases [16]. To date, overall performance in identifying different emotional prosodies generally varies between 40% and 99.5% depending on the classification model [17].

The most frequent speech features used to identify affective prosodies in speech are presented in Table 1 [9,16,18–22].

**Table 1.** Preferred features for speech emotion recognition.

| Type | Feature | Description | Interpretation |
|---|---|---|---|
| Prosodic | Fundamental frequency or pitch (Hertz) | Minimal frequency of a period. | High or low frequency perception. |
| | Speech rate | Number of syllables per unit of time. | Index of speaker arousal. |
| | Intensity (dB) and Energy (Volts) | Mean intensity Root mean square energy. | Loudness |
| Voice quality | Jitter (%) | Index of frequency variations from period to period. | Pitch fluctuations. |
| | Shimmer (%) | Index of amplitude variations from period to period. | Loudness fluctuations. |
| | Harmonics-to-noise ratio (dB) | Relation of the energy of harmonics against the energy of noise-like frequencies. | Voice breathiness and roughness. |
| Spectral | Formants (Hertz) | Frequencies of highest energy. | Amplification of frequencies caused by resonances in the vocal tract. |
| | Mel Frequency Cepstral Coefficients (MFCC) | Discrete cosine transform of the log power spectrum filtered by a Mel-filter bank. | Representation of spectral information according to the human auditory frequency response. |

Prosodic features have been broadly utilized as they allow a direct distinction between emotional states. For instance, when compared to neutral speech, the fundamental frequency tends to be higher than average with higher fluctuations and wider frequency ranges for angry, happy and afraid prosodies [23,24]. Voice quality features are perceptual correlates of the voice profile (e.g., harsh, tense, breathy) and may depend on the speaker's emotional state. Specifically, the modulations of muscular activities of thoracic, abdominal, laryngeal and orofacial muscles elicited by the emotional induction impact of both the subglottal pressure and the expiratory airflow from the lungs, triggering irregular periodicity of glottal pulses. In situations of high arousal, for instance, the voice may be

perceived as rougher (as indexed by lower harmonic-to-noise ratio), and higher short-term period-to-period variability of fundamental frequency and amplitude of glottal pulses (i.e., respectively, jitter and shimmer) are expected [25].

The vocal tract acts as a group of filters that shape the acoustic waveform from the larynx to the pharyngeal, oral, and nasal cavities. The air pressure controlled by the respiratory muscles triggers the vibration of the vocal folds that produces a periodic complex waveform and results in voiced signals. Once pressure fluctuations from the closed end of the vocal tract (glottis) reach the open end (lips), the acoustic waveform reflects with an inverse polarity back onto the vocal tract because it strikes with a mass of air particles. The time for the waveform to travel from one extremity of the vocal tract to the other determines the resonance (or formant) frequency. The longer the distance is, the higher the amount of time taken, and the vocal tract resonates for waves that have longer periods, and thus lower frequencies [26]. Therefore, resonance frequencies directly depend on the length of the vocal tract, which are shaped by neuro-muscular modulations of vocal organs during emotional perception. For instance, happy prosody was characterized by shorter vocal tract length than angry and sad speech [27]. Finally, MFCC are frequently useful for emotional discrimination and allow us to optimize the spectral representation of speech according to the human auditory frequency response using the logarithmic Mel-Scale [28]. Nevertheless, the distribution of the weight of specific acoustic features to distinguish between emotions is convoluted [29]. For instance, fundamental frequency, speaking rate and energy are bad features to recognize anger versus happiness as compared to the neutral state, as both angry and happy prosodies are characterized by higher average pitch with a wider spectral range, higher loudness, and faster speech. However, these acoustic features are useful to highlight fear versus sadness prosodical patterns as compared to neutral speech [23]. Therefore, features must be combined and accurately modeled to depict the most pertinent information for emotional characterization [29].

Many predictive models have been trained on the basis of spectral, prosodic and voice quality features to recognize emotional patterns. Typical machine learning algorithms stand on five models: (1) Support Vector Machines (SVM), (2) Hidden Markov Model, (3) k-Nearest Neighbors, (4) Decision Trees and (5) Artificial Neural Networks. Unfortunately, the classification method has not been standardized yet [17]. The selection of classification algorithms mostly depends on the data properties (e.g., number of observations and classes) and the nature and number of features (the attributes of the object to be identified).

In light of the above discussion, the present work aims to contribute to the growing research concerning speech emotion recognition by providing a Mexican Emotional Speech Database (MESD). For this purpose, a database containing Mexican-shaped emotional utterances for child, female and male voices was first created. Words for emotional utterances come from two corpora: (corpus A) is composed of 24 nouns and adjectives coming from the INTERFACE for Castilian Spanish database [30] and repeated across emotional prosodies and types of voice (female, male and child), and (corpus B), which consists of 24 words per emotional prosody controlled for age of acquisition, frequency of use, familiarity, concreteness, valence, arousal and discrete emotion dimensionality ratings. See Figure 1 for details about the MESD content. The MESD contains 288 single-word utterances per type of voice (i.e., adult male, adult female and child) which have been previously selected by supervised learning means in order to guarantee the optimal emotional distinction between anger disgust, fear, happiness, neutral and sadness prosodies (48 utterances each: 24 per word corpus). Specifically, 1152 emotional utterances per type of voice (i.e., 192 single words per emotion: 96 per word corpus) were recorded and a method for selecting utterances that best represent each emotional prosody was proposed. Particularly, a cubic SVM model was accordingly trained and validated on single units of the whole recording repository that contain 48 utterances (24 from corpus A and 24 from corpus B) per emotion (i.e., 288 single words). This procedure was repeated for each type of voice. The performance for emotional recognition on each unit was considered as an index of the utterance's validity for emotional content. This way, the most representative utterances

of each emotional prosody were selected to finally integrate the MESD. Hereinafter, this evaluation method will be referred to as the SVM-based validation method. The last step to confirm the MESD emotional representativity was to evaluate the performance of the cubic SVM model to recognize the six emotional prosodies independently for male, female, and child voices.

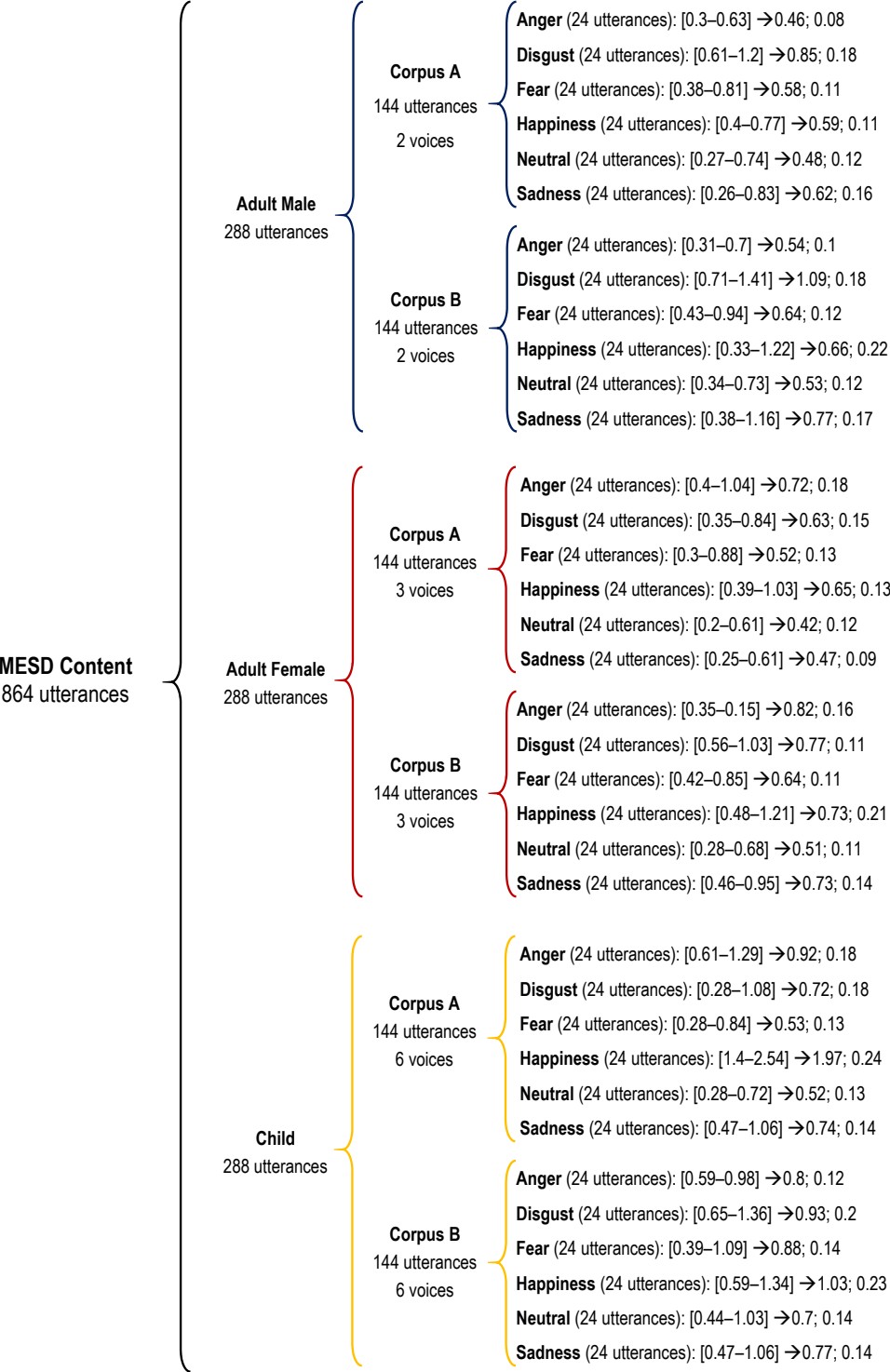

**Figure 1.** Mexican Emotional Speech Database (MESD) content. Numbers represent information about single-word utterance durations, expressed in seconds. Bracketed values are from left to right: minimum and maximum durations; the third value is the mean duration and the last is the standard deviation across 24 utterances.

It is important to note that, consequently, the supervised learning study in this work was not designed to optimize performances for emotion recognition on the whole recording repository, but to identify a small fragment of it (a 288-uterrance unit) that best represents the prosodic discrimination between emotions. The main objective of this study is to guarantee the ability for future listeners to differentiate emotional prosodies. Therefore, the cubic SVM model was used as tool for objectively creating the database.

We chose an SVM model to assess the validity of emotional content because of its effectiveness to compute small datasets (as compared to other algorithms such as deep neural networks) and its ability for handling high-dimensional spaces [31]. Recently, Acilin and Milton extracted spectral features of speech recordings from six databases: the Berlin Database of Emotional Speech (German), the Ravdess database (North American English), the Savee database (British English), the EMOVO emotional speech database (Italian), the eNTERFACE database (English), and the Urdu database (Urdu) [28]. A multiclass SVM classifier was used to recognize emotions in a one-against-one approach. Using MFCC as the input features, overall accuracies reached up to 73.64% on the Berlin database, 31.25% on the Ravdess, 69.17% on the Savee, 53.40% on the EMOVO, 48.33% on the eNTERFACE, and 85.75% on the Urdu database. SVM performance was increased by at most 32.3% if considering Mel frequency magnitude coefficients. In another study, the Berlin Database of Emotional Speech was used to compare performances for emotion recognition between Back Propagation Neural Networks, Extreme Learning Machine, Probabilistic Neural Network, and SVM supervised learning models based on MFCC [32]. The SVM classifier outperformed all the other three models by reaching 92.4% overall accuracy compared to 77.8%, 77.4% and 81% for the Back Propagation Neural Networks, Extreme Learning Machine and Probabilistic Neural Network, respectively. In a multi-modal approach (emotional recognition based on prosodic, spectral, voice quality acoustic features and text parameters), SVM has been used as a late fusion algorithm to recollect previously obtained prediction information about emotion recognition, based on acoustic and text features separately, and generate a final multi-modal predictive model. The authors observed that the late SVM-based fusion improved the performance for emotion recognition [33].

In the present study, we propose a cubic SVM model to assess the validity of the emotional content of (1) the 1152 single-word-utterance initial repertory from each type of voice (male, female, and child), and (2) the 288-utterance dataset per type of voice finally integrated into the MESD. Subsequently, the proposed cubic SVM model was applied on the previously validated INTERFACE database to assess the MESD reliability. Additionally, a statistical evaluation was undertaken to explore cofounding effects of linguistic parameters (words frequency of use, familiarity, and concreteness) on affective prosodies. Finally, culturally specific patterns are highlighted and discussed.

## 2. Results

### 2.1. MESD Speech Corpus: Corpus B

No inter-emotion difference was emphasized for frequency of use, familiarity, and concreteness ratings after outliers were removed. For male, female and child databases, statistical analysis stressed non-significant *p*-values ($p > 0.05$) for each parameter.

### 2.2. SVM-Based Validation Method

#### 2.2.1. Female Adult Voice

Figure 2 details the SVM validation performance on data from each female participant individually. Confusion matrices are presented. Table 2 details the F-scores for each participant. Each actor uttered 288 single words, of which 77% (222 utterances) were used for training, and 23% (66 utterances) for validation. Hereafter, "Pn" will be used to refer to participant number n.

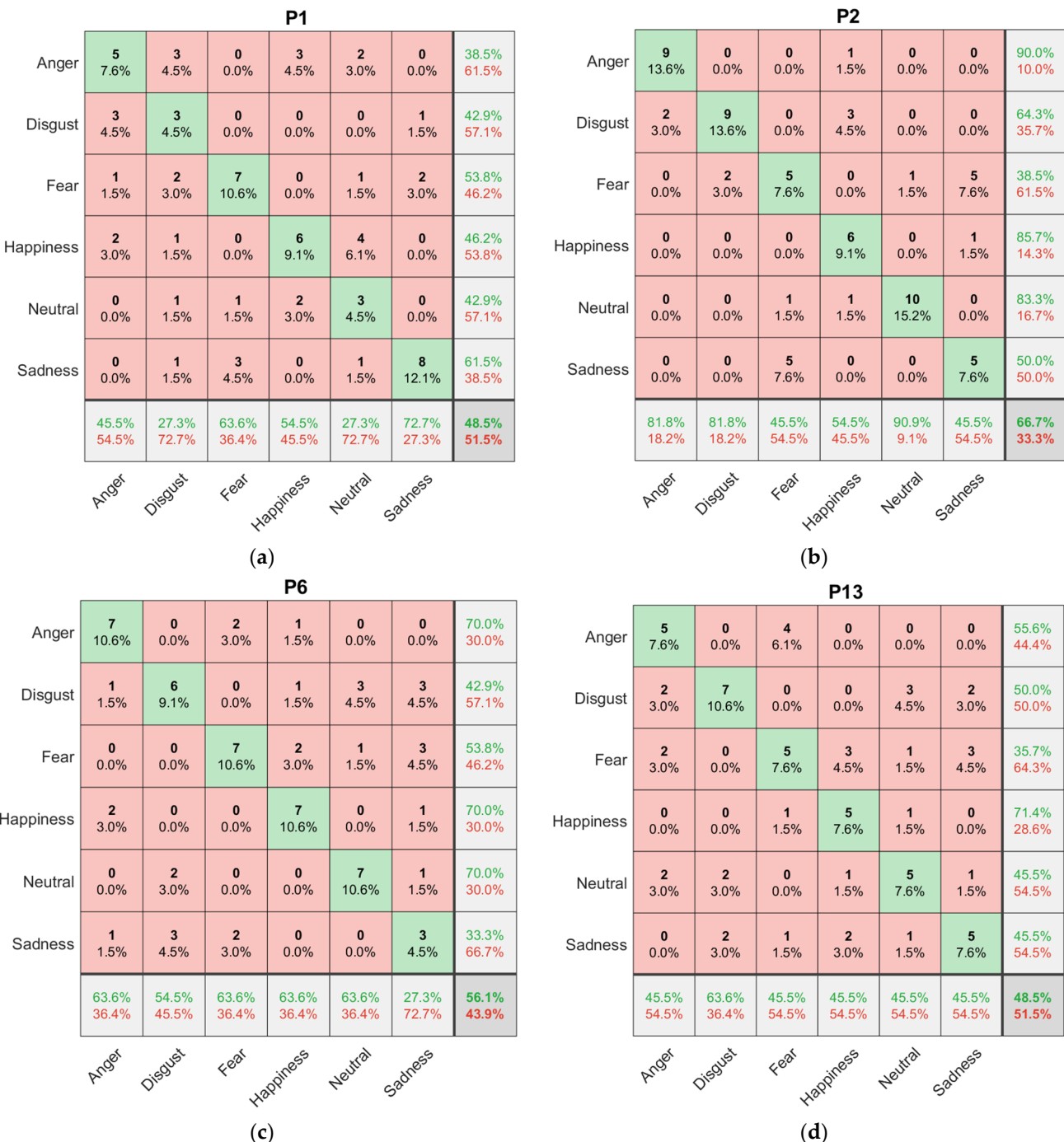

**Figure 2.** Single-actor SVM performance (confusion matrices) on female voices. Rows present the predicted emotion, and columns the targeted emotion. Diagonal cells in green correspond to observations that were correctly classified. Other cells in red correspond to incorrectly classified observations. Each cell includes both the number of observations and the percentage of the total number of observations. The column on the far right states the precision (in green) and the false discovery rate (in red), both expressed in percentage. The row at the bottom of the plot details the recall (in green) and the false negative rate (in red), both expressed in percentage. The cell on the bottom far right of the plot details the overall accuracy. (**a**) P1, (**b**) P2, (**c**) P6, (**d**) P13.

Figure 3 presents the SVM validation performance (confusion matrices) on data from female voices of the final version of MESD, in comparison with the outcomes reported for female voices of INTERFACE. Table 3 details corresponding F-scores. It is important to note that the most representative female participants for each emotion resulting from the single-actor classification were: (1) P2 for anger, (2) P2 for disgust, (3) P1 for fear, (4) P6 for

happiness, (5) P2 for neutral and (6) P1 for sadness. See Table 2 for F-scores obtained after the SVM classification process conducted individually on the data from those actors.

**Table 2.** Single-actor F-score (%) from emotion classification based on female and male individual datasets.

| Participant | Anger | Disgust | Fear | Happiness | Neutral | Sadness | Total |
|---|---|---|---|---|---|---|---|
| | | | Female | | | | |
| P1 | 41.7 | 33.3 | 58.3 | 50 | 33.3 | 66.7 | 47.2 |
| P2 | 85.7 | 72 | 41.7 | 66.7 | 87 | 47.6 | 66.8 |
| P6 | 66.7 | 48 | 58.3 | 66.7 | 66.7 | 30 | 56 |
| P13 | 50 | 56 | 40 | 55.6 | 45.5 | 45.5 | 48.7 |
| | | | Male | | | | |
| P3 | 95.2 | 90.9 | 70 | 84.6 | 100 | 76.2 | 86.1 |
| P8 | 53.3 | 46.1 | 40 | 44.4 | 55.6 | 28.6 | 44.7 |
| P12 | 84.2 | 91.7 | 69.6 | 80 | 84.2 | 90.9 | 83.4 |
| P14 | 63.1 | 37.5 | 53.8 | 38.5 | 36.4 | 52.2 | 46.9 |

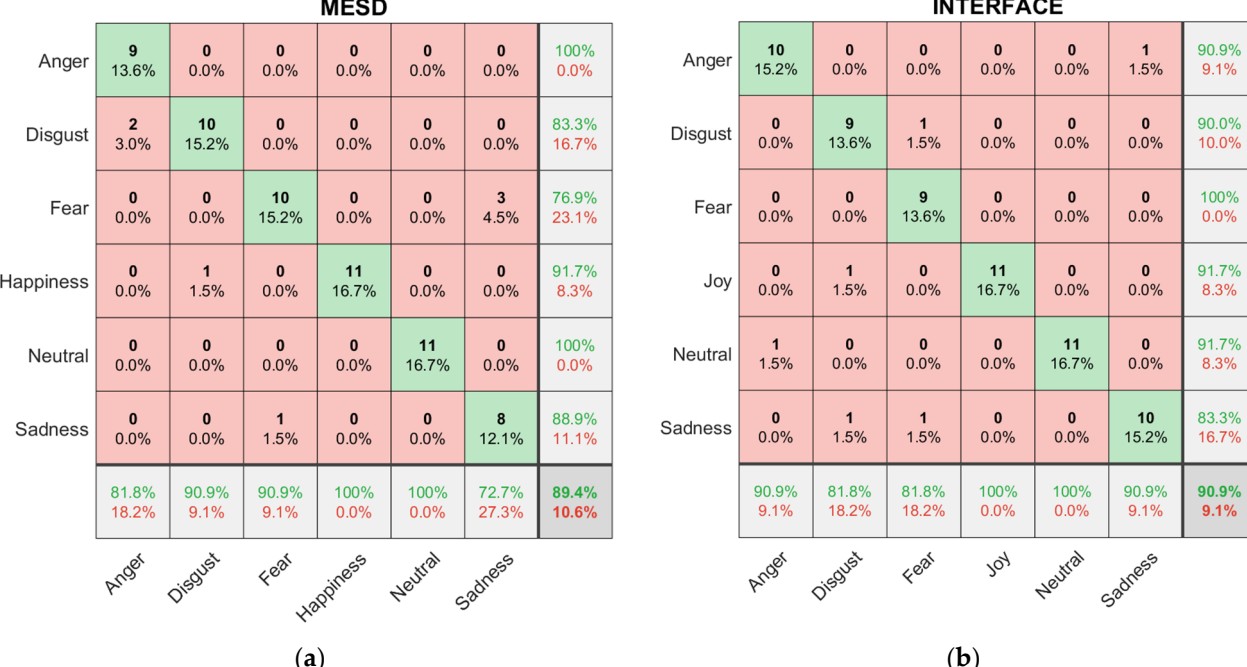

(**a**)     (**b**)

**Figure 3.** SVM performance (confusion matrices) on female voices for (**a**) MESD and (**b**) INTERFACE database. Rows present the predicted emotion and columns the targeted emotion. Diagonal cells in green correspond to observations that were correctly classified. Other cells in red correspond to incorrectly classified observations. Each cell includes both the number of observations and the percentage of the total number of observations. The column on the far right states the precision (in green), and the false discovery rate (in red), both expressed in percentage. The row at the bottom of the plot details the recall (in green) and the false negative rate (in red), both expressed in percentage. The cell on the bottom far right of the plot details the overall accuracy.

**Table 3.** F-score (%) from emotion classification on female voices from MESD and INTERFACE.

| Database | Anger | Disgust | Fear | Happiness | Neutral | Sadness | Total |
|---|---|---|---|---|---|---|---|
| MESD | 90 | 87 | 83.3 | 95.7 | 100 | 80 | 89.3 |
| INTERFACE | 90.9 | 85.7 | 90 | 95.7 | 95.7 | 87 | 90.9 |

### 2.2.2. Male Adult Voice

Figure 4 details the SVM validation performance on data from each male participant individually. Confusion matrices are presented. Table 2 details the F-scores for each participant. Each actor uttered 288 single words of which 77% (222 utterances) were used for training, and 23% (66 utterances) for validation.

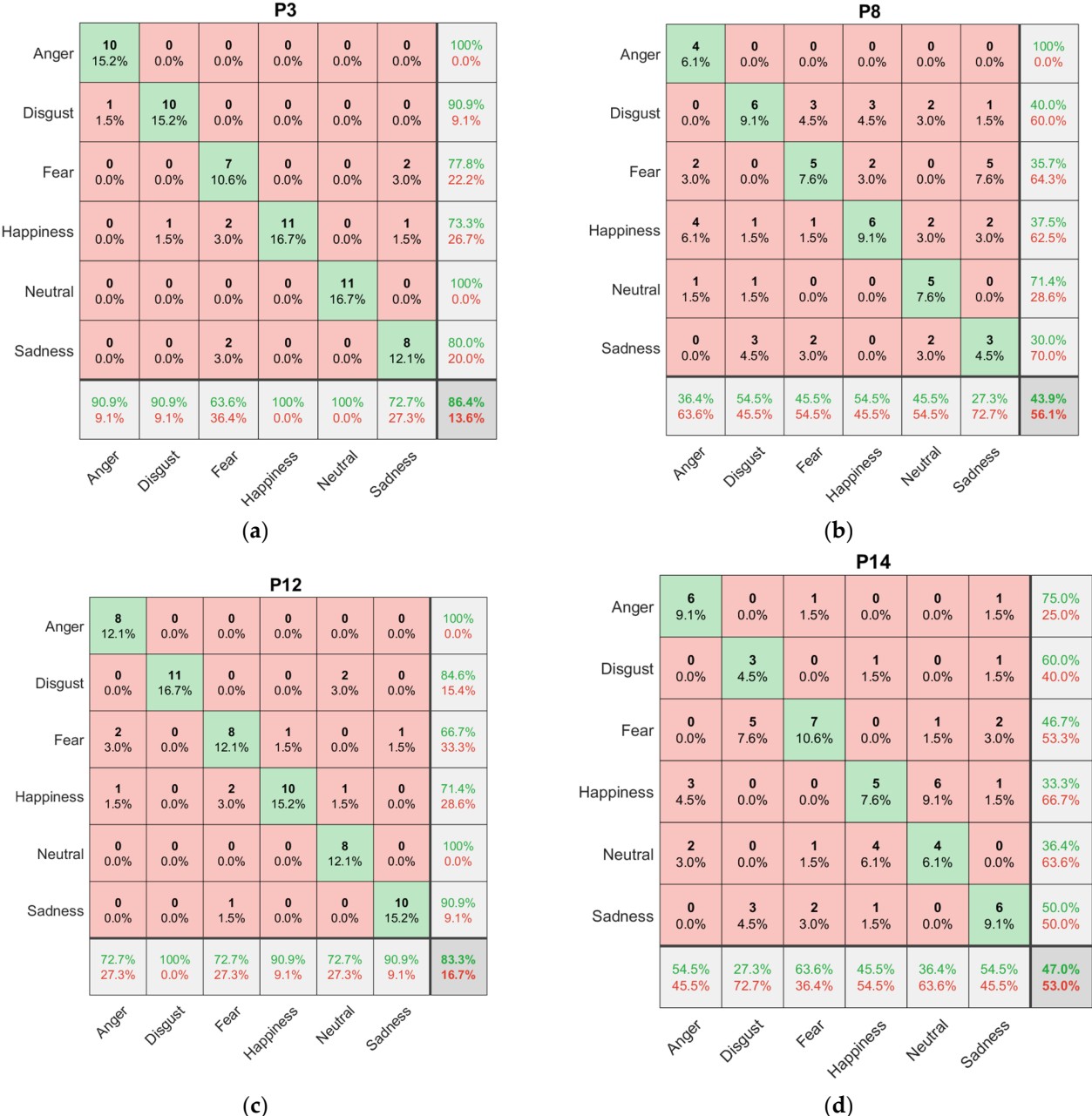

**Figure 4.** Single-actor SVM performance (confusion matrices) on male voices. Rows present the predicted emotion, and columns the targeted emotion. Diagonal cells in green correspond to observations that were correctly classified. Other cells in red correspond to incorrectly classified observations. Each cell includes both the number of observations and the percentage of the total number of observations. The column on the far right states the precision (in green), and the false discovery rate (in red), both expressed in percentage. The row at the bottom of the plot details the recall (in green) and the false negative rate (in red), both expressed in percentage. The cell on the bottom far right of the plot details the overall accuracy. (**a**) P3, (**b**) P8, (**c**) P12, (**d**) P14.

Figure 5 illustrates the SVM validation performance (confusion matrices) on data from male voices of the final version of MESD, in comparison with the outcomes reported for male voices of INTERFACE. Table 4 details the corresponding F-scores. Note that the most representative male participants for each emotion resulting from the individual classification were: (1) P3 for anger, (2) P12 for disgust, (3) P3 for fear, (4) P3 for happiness, (5) P3 for neutral, and (6) P12 for sadness. See Table 2 for accuracy and F-scores obtained after the SVM classification process conducted individually on the data from those actors.

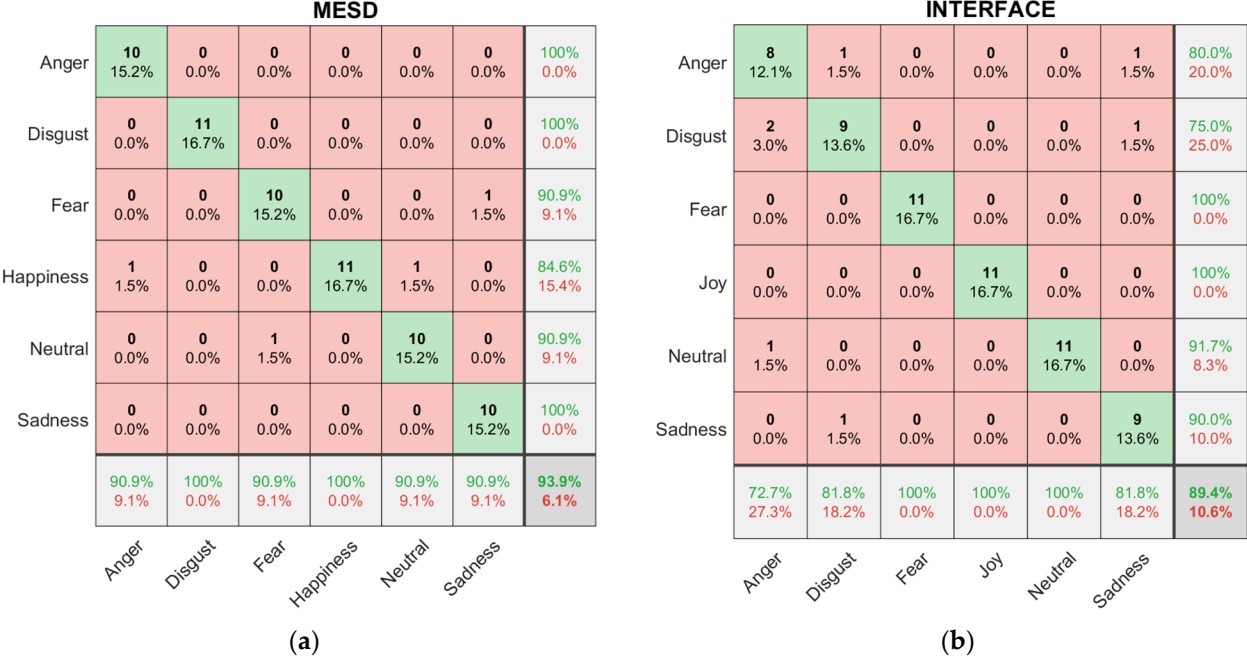

(**a**)                             (**b**)

**Figure 5.** SVM performance (confusion matrices) on male voices for (**a**) MESD and (**b**) INTERFACE database. Rows present the predicted emotion, and columns the targeted emotion. Diagonal cells in green correspond to observations that were correctly classified. Other cells in red correspond to incorrectly classified observations. Each cell includes both the number of observations and the percentage of the total number of observations. The column on the far right states the precision (in green), and the false discovery rate (in red), both expressed in percentage. The row at the bottom of the plot details the recall (in green) and the false negative rate (in red), both expressed in percentage. The cell on the bottom far right of the plot details the overall accuracy.

**Table 4.** F-score (%) from emotion classification on male voices from MESD and INTERFACE.

| Database | Anger | Disgust | Fear | Happiness | Neutral | Sadness | Total |
|---|---|---|---|---|---|---|---|
| MESD | 95.2 | 100 | 90.9 | 91.7 | 90.9 | 95.2 | 94 |
| INTERFACE | 76.2 | 78.3 | 100 | 100 | 95.7 | 85.9 | 89.3 |

### 2.2.3. Child Voice

Figure 6 summarizes the number of observations for single actors in both clusters after k-means clustering for utterances from each emotion separately. Consequent pairs of participants are detailed in Table 5.

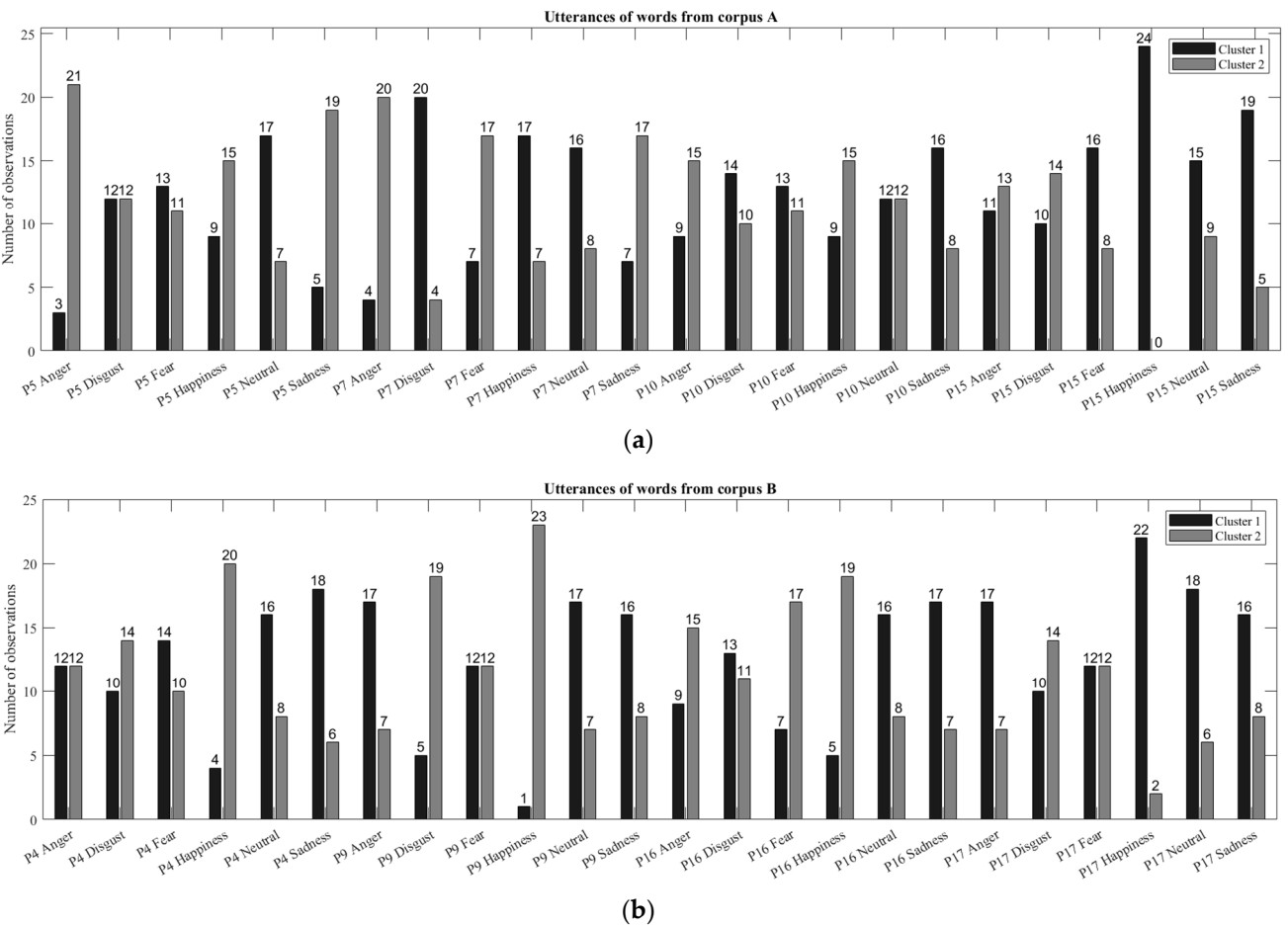

**Figure 6.** Representativity of single actor utterances in both clusters for anger, disgust, fear, happiness, neutral, and sadness prosodies: utterances of words from corpus A (**a**) and words from corpus B (**b**).

**Table 5.** Corpus A/Corpus B child actor pairs and F-score (%) from emotion classification.

| Actor Pairs Resulting from Clustering | | Single-Pair F-Score (%) from Emotion Classification | | | | | | |
|---|---|---|---|---|---|---|---|---|
| Index | Pair | | | | | | | |
| | Participant Who Uttered Words from Corpus A | Participant Who Uttered Words from Corpus B | Anger | Disgust | Fear | Happiness | Neutral | Sadness | Total |
| 1 | P15 | P9 | 50 | 66.7 | 40 | 60 | 63.6 | 28.6 | 51.5 |
| 2 | P15 | P17 | 36.4 | 46.2 | 27.3 | 90 | 38.1 | 28.6 | 44.4 |
| 3 | P5 | P16 | 80 | 52.2 | 78.3 | 63.7 | 36.4 | 81.2 | 59.2 |
| 4 | P7 | P16 | 60 | 52.2 | 48 | 47.7 | 66.7 | 54.6 | 54.9 |
| 5 | P15 | P4 | 32 | 25 | 37 | 66.6 | 37.5 | 36.4 | 39.1 |
| 6 | P10 | P9 | 30.8 | 34.8 | 56 | 22.2 | 63.2 | 47.7 | 42.5 |
| 7 | P5 | P17 | 43.5 | 42.1 | 33.3 | 40 | 46.2 | 64 | 44.9 |
| 8 | P10 | P4 | 32.3 | 10.5 | 09.5 | 52.2 | 44.4 | 30 | 29.9 |
| 9 | P5 | P9 | 66.7 | 55.2 | 57.1 | 70 | 33.3 | 69.6 | 58.7 |
| 10 | P10 | P16 | 63.7 | 52.2 | 74.1 | 11.8 | 27.3 | 38.1 | 41.9 |

Figure 7 details the SVM validation performance on data from each single pair of child participants individually. Confusion matrices are presented. Table 5 details the F-scores for each pair of actors. Each pair corresponds to 288 utterances, of which 77% (222 utterances) were used for training, and 23% (66 utterances) for validation.

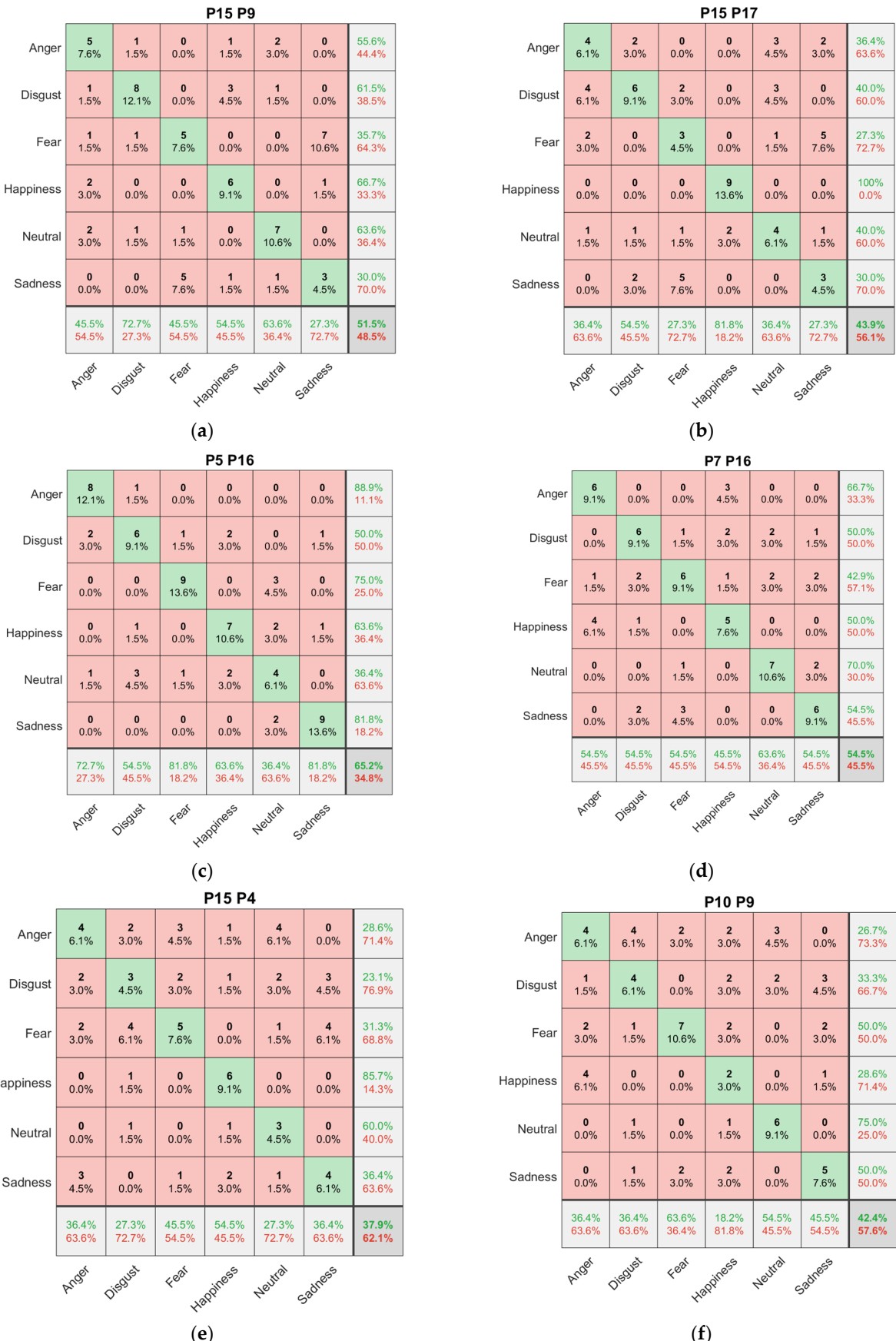

**Figure 7.** *Cont.*

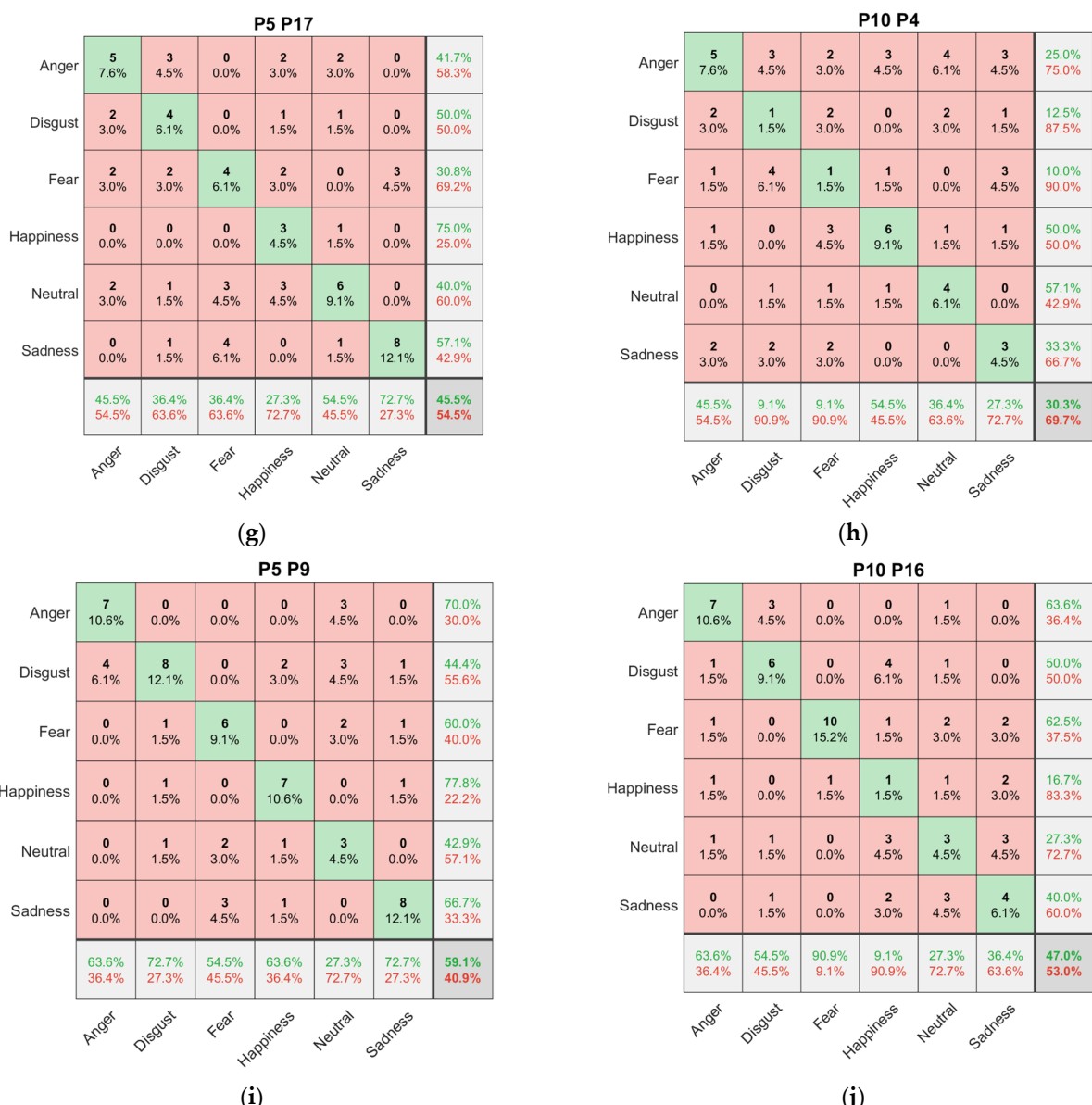

**Figure 7.** Single-pair SVM performance (confusion matrices) on child voices. Rows present the predicted emotion, and columns the targeted emotion. Diagonal cells in green correspond to observations that were correctly classified. Other cells in red correspond to incorrectly classified observations. Each cell includes both the number of observations and the percentage of the total number of observations. The column on the far right states the precision (in green), and the false discovery rate (in red), both expressed in percentage. The row at the bottom of the plot details the recall (in green) and the false negative rate (in red), both expressed in percentage. The cell on the bottom far right of the plot details the overall accuracy. (**a**) Pair 1, (**b**) Pair 2, (**c**) Pair 3, (**d**) Pair 4, (**e**) Pair 5, (**f**) Pair 6, (**g**) Pair 7, (**h**) Pair 8, (**i**) Pair 9, (**j**) Pair 10.

Figure 8 illustrates the SVM validation performance (confusion matrices) on data from child voices of the final version of MESD. Table 6 details the corresponding F-scores. Note the most representative actor pairs for each emotion resulting from the single-pair classification were: (1) pair 3 for anger, (2) pair 1 for disgust, (3) pair 3 for fear, (4) pair 2 for happiness, (5) pair 4 for neutral, and (6) pair 3 for sadness. See Table 5 for accuracy and F-scores obtained after the SVM classification process conducted individually on the data from these pairs of actors.

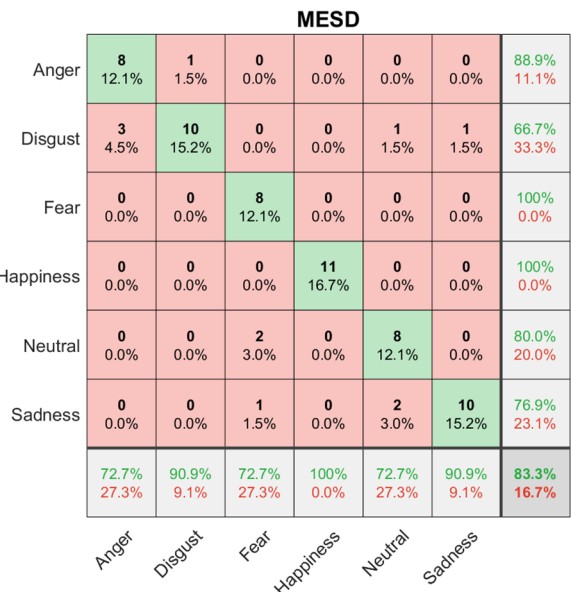

**Figure 8.** SVM performance (confusion matrix) on child voices for MESD. Rows present the predicted emotion, and columns the targeted emotion. Diagonal cells in green correspond to observations that were correctly classified. Other cells in red correspond to incorrectly classified observations. Each cell includes both the number of observations and the percentage of the total number of observations. The column on the far right states the precision (in green), and the false discovery rate (in red), both expressed in percentage. The row at the bottom of the plot details the recall (in green) and the false negative rate (in red), both expressed in percentage. The cell on the bottom far right of the plot details the overall accuracy.

**Table 6.** F-score (%) from emotion classification on child voices from MESD.

| Database | Anger | Disgust | Fear | Happiness | Neutral | Sadness | Total |
|---|---|---|---|---|---|---|---|
| MESD | 80 | 76.9 | 84.2 | 100 | 76.2 | 83.3 | 83.4 |

*2.3. Statistical Analysis*

2.3.1. Effects of Emotions and Cultures

Significant emotion effects ($p < 0.05$) on acoustic features were highlighted for both adult (male and female) and child voices of MESD and INTERFACE utterances. See Table 7 for significant trends in Mexican and Castilian Spanish independently for adult male, female and child voices.

Interestingly, inter-culture comparisons showed a significant cultural shaping of emotional prosodies on both adult male and adult female voices. Table 8 details significant inter-culture trends on acoustic features for both female and male voices. As expected, prosodic, spectral and voice quality features were modulated by culture. Furthermore, all emotions were affected by cultural shaping.

2.3.2. Effects of Words Familiarity, Frequency and Concreteness

MESD utterances of words from corpus A and words from corpus B were compared, and differences were assessed for each emotion. A significant effect was highlighted for both adult (male and female) and child voices on prosodic, spectral and voice quality features. Furthermore, word frequency, familiarity and concreteness affected all emotional prosodies (anger, disgust, fear, happiness, neutral and sadness). See Table 9 for significant trends on acoustic features for adult female, adult male and child voices.

**Table 7.** Effect of emotion on acoustic features.

| Acoustic Feature | Significant Trends | | | | |
|---|---|---|---|---|---|
| | Female | | Male | | Child |
| | INTERFACE | MESD | INTERFACE | MESD | MESD |
| Rate | J[1] > D[2], J > S[3], N[4] > S, A > F/N, A > D/S, J > F/N, F/N > D | N > A[5], F[6] > D, N > D, F > H[7], N > F, N > H, S > H, N > S, F/S > A, F/S > H | A > D, N > A, S > D, F/J > A, J/N/F > D, J/N/F > S | A > F/H, S > D, N > F, F > S, N > H, A/N > D, F/H > D, A/N > S | N > H, N > A/D/S, A/D/F/S > H |
| Mean pitch | A > D, J > A, A > N, F > D, J > D, D > N, J > F, F > N, F > S, J > N, J > S, S > N | H > A, D > N, H > N, S > N, D/F/S > A, H > F/S/D, F > N | F > A, A > S, F > D, F > J, F > N, F > S, J > S, A/J > D/N, D/N > S | N > D, A/H > D, F/S > D, A/H > N, F/S > N, F/S > H, F/S > A | H > D/N, H > A/S, A/F/S > D/N |
| SD pitch | A > N, J > N, J > S, A > D/F/S, D/F/S > N | D/H > A, A > N/S, D/H > F/N/S | D > A, D > F, D > J, A/F/J > N/S | A > N, D/F/H/S > N, A > D/F/H/S | A/D/H > F/N/S |
| Jitter local | A > J, D > F, N > J, N > D/S, N > F/A, D/S > J | F/N > A/D/H/S | S > D, J > F, S > F, S > J, S > N, D/N > A/F | A > F, H > S, D > A/N, D > H, D > S/F, A/N > S | A > H, F > H, D > A/F/N/S, N/S > H |
| Jitter ppq5 | A > J, N > A, S > F, N/S > J, D/F > J, D/S > A, N > F/D | D/S > A, F/N > A, F/N > D/S, D/S > H, F/N > H | D > A, S > A, D > F, D > J, S > D, N > F, S > F, S > J, S > N | N > S, A > F/H/S, D > F/H/S | A > H, D > H, D > S, F/N/S > H, D > A/F |
| Shimmer local | A/N > F | S > A, F > H, F > S, F/N > A/D, S/N > H | A > F, A > N, D > F, J > F, S > F, D/S > A/J, D/S > N | A > D, A > F, A > H, A > N, A > S, D > H, D/F > S, N > H, H > S, N > S | N > A, N > H, N > S, A/D/F/S > H |
| Shimmer rapq5 | A > F | F > S, F/N > D, F/N > A/H, D/S > A/H | S > F, A/D > F, D/S > J/N | A > S, A > D/F/H/N, D/F/H/N > S | N > S, A/D/F/N > H |
| Mean HNR | F/S > A, J/N/D > A, F/S > D, F/S > J, F/S > N | N/S/D/A > F/H | F > A, F > D, S > D, F > J, F > N, F > S, J/N > A/D | D > A, F > A, H > A, N > A, S > A, F > D, H > D, N > D, S > D, F > H, F > S, N > S, F, N > F, N > H, S > H, S > N | S > D, A/F/H/N > D, S > A/F/H/N |
| Mean intensity | F > N, A/D/J/S > N, F > D/S/J | A > F, A > S, D > S, F > S, A > D/H/N, H/N/D > F, H/N > S | F > A, J > A, F > D, J > D, D > S, D/F > J, F > N, F > S, J > N, J > S, N > S, A/N > D | A > D, A > H, A > N, H > D, N > D, H > F, H > N, H > S, A > S/F, S/F > D, S/F > N | A > D, H > A, H > D, D > N/S, H > N/F/S, A > N/F/S |
| SD intensity | None | None | S > D, S > N, A/S > J/F | S > A/D | None |
| F1 * | D > S | F/H/S > A/D/N | A/D > F | A > S, A > D/F/H/N, D/H/N > S | H/D/F > N/S |
| F2 * | None | F/H/S > A/D, H/S > N | None | D/H > F, D > N, S > A, | A/D/F/N > H |
| F3 * | D > A/F/J | H > F | D > S, D/F > A/J/N | A > N, D > N, S > F, S > H, S > N, D > A/F/H | D > A/F/N/S, A/F/N/S > H |
| B1 † | None | None | A/S > N/J/F | A > D/S, H > S | D > N/S |
| B2 † | J > N, F/S > J, F/S > A | None | S > J | None | None |
| B3 † | F/N > A, F/N > J | None | F/N > A/J | None | A > H, D > F/H/N/S |
| RMS + Energy | F > D, F > N, A > D/J, A > N/S, D/J > N, J/F > S | A > D/H/N, A > F/S, S > F/S, H/N > F/S | A > D, F > A, J > A, F > D, J > D, F > J, F > N, F > S, J > N, J > S, N/S > D | A > H, H > D, H > F, H > N, H > S, F/S > D, A > D/N, A > F/S, F/S > N | A > D, H > A, A > N, H > D, D > N, F > N, H > N, S > N |

[1] Joy, [2] Disgust, [3] Sadness, [4] Neutral, [5] Anger, [6] Fear, [7] Happiness, * F1, F2, F3 = Formants 1, 2, 3, respectively, † B1, B2, B3 = Formant bandwidth in center (F1, F2, F3, respectively), + Root Mean Square Energy.

**Table 8.** Effect of culture on acoustic features extracted from emotional utterances.

| Acoustic Feature | Inter-Culture (Mexican versus Spanish) Significant Trends | |
|---|---|---|
| | Female | Male |
| Rate | I[1] (A[2]) > M[3] (A), M (S[4]) > I (S) | I (F) > M(F) |
| Mean pitch | I (A) > M (A), M (D[5]) > I (D), I (F[6]) > M (F), M (N[7]) > I (N) | M (S) > I (S) |
| SD pitch | M (D) > I(D), M (H[8]) > I (J[9]), M (N) > I (N) | M (A) > I (A), M (H) > I (J), I (N) > M (N), M (S) > I (S) |
| Jitter local | None | I (S) > M (S) |
| Jitter ppq5 | None | I (F) > M (F), I (S) > M (S) |
| Shimmer local | I (A) > M (A), M (F) > I (F), I (J) > M (H) | I (S) > M (S) |
| Shimmer rapq5 | I (A) > M (A) | M (A) > I (A), M (N) > I (N) |
| Mean HNR | M (A) > I (A), I (F) > M (F) | I (A) > M (A), M (N) > I (N), M (S) > I (S) |
| Mean intensity | M (A) > I (A), M (D) > I (D), I (F) > M (F), M (H) > I (J), M (N) > I (N) | M (A) > I (A), I (D) > M (D), I (F) > M (F), I (J) > M (H), I (N) > M (N) |
| SD intensity | None | None |
| F1 * | M (S) > I (S) | None |
| F2 * | None | None |
| F3 * | M (D) > I (D), M (H) > I (J) | M (S) > I (S) |
| B1 † | None | M (A) > I (A), M (D) > I (D), M (F) > I (F), M (H) > I (J), M (N) > I (N), M (S) > I (S) |
| B2 † | None | None |
| B3 † | I (A) > M (A), I (D) > M (D), I (F) > M (F), I (N) > M (N), I (S) > M (S) | None |
| RMS + Energy | I (F) > M (F), I (S) > M (S) | I (D) > M (D), I (F) > M (F), I (J) > M (H), I (N) > M (N), I (S) > M (S) |

[1] INTERFACE database, [2] Anger, [3] MESD, [4] Sadness, [5] Disgust, [6] Fear, [7] Neutral, [8] Happiness, [9] Joy, * F1, F2, F3 = Formants 1, 2, 3, respectively, † B1, B2, B3 = Formant bandwidth in center (F1, F2, F3, respectively), + Root Mean Square Energy.

**Table 9.** Effect of word familiarity, frequency and concreteness on acoustic features.

| Acoustic Feature | Inter-Corpus (Corpus A versus Corpus B) Significant Trends | | |
| --- | --- | --- | --- |
| | **Female** | **Male** | **Child** |
| Rate | None | None | M [1] (A [2]) > I [3] (A), M (H [4]) > I (H) |
| Mean pitch | I (A) > M (A), I (D [5]) > M (D), I (N) > M (N), | None | M (A) > I (A), M (F [6]) > I(F), M(H) > I(H), M (N [7]) > I (N), M (S [8]) > I (S) |
| SD pitch | I (D) > M (D) | None | M (A) > I (A), M (D) > I (D), M (F) > I (F), M (N) > I (N), M (S) > I (S) |
| Jitter local | None | M (F) > I (F) | I (F) > I (M), M (H) > I (H) |
| Jitter ppq5 | None | M (F) > I (F) | M (D) > I (D), M (H) > I (H) |
| Shimmer local | M (A) > I (A) | None | M (A) > I (A), M (H) > I (H), I (S) > M (S) |
| Shimmer rapq5 | M (A) > I (A) | None | M (A) > I (A), M (H) > I (H), M (N) > I (N) |
| Mean HNR | None | I (F) > M (F), I (H) > M (H), I (N) > M (N) | M (F) > I (F), I (H) > M (H) |
| Mean intensity | I (S) > M (S) | M (F) > I (F), M (H) > I (H), M (N) > I (N) | None |
| SD intensity | None | None | None |
| F1 * | None | None | M (H) > I (H) |
| F2 * | None | M (A) > I (A), M (F) > I (F), M (H) > I (H) | M (S) > I (S) |
| F3 * | M (A) > I (A), M (H) > I (H) | M (A) > I (A), M (D) > I (D) | M (D) > I (D), M (H) > I (H), M (S) > I (S) |
| B1 † | None | None | I (N) > I (N) |
| B2 † | None | None | I (F) > I (F), I (H) > I (H) |
| B3 † | M (A) > I (A), M (F) > I (F), M (N) > I (N) | None | None |
| RMS + Energy | None | M (F) > I (F), M (H) > I (H), M (N) > I (N), I (S) > M (S) | I (D) > M (D), I (F) > M (F) |

[1] word from corpus B, [2] Anger, [3] words from corpus A, [4] Happiness, [5] Disgust, [6] Fear, [7] Neutral, [8] Sadness, * F1, F2, F3 = Formants 1, 2, 3, respectively, † B1, B2, B3 = Formant bandwidth in center (F1, F2, F3, respectively), + Root Mean Square Energy.

## 3. Discussion

In the present work, a new database is introduced: the Mexican Emotional Speech Database (MESD) that contains single-word emotional expressions uttered by children and adult males and females. This database was validated by applying an SVM-based method and comparing it with the INTERFACE database when possible. In addition, a statistical evaluation of the MESD was undertaken to analyze the effects of emotion, culture and language (word frequency of use, concreteness, and familiarity) on acoustic features that define emotional prosodies. The process validation outcome is hereunder discussed.

### 3.1. SVM-Based Validation Process

The validation process led to competitive performances for speech emotion recognition when MESD and INTERFACE were compared. As a reference, Table 10 shows performances reached in previous investigations for emotion pattern prediction in speech.

The present work proposes the SVM-validation method that yielded 89.4% for male and 90.9% for female voice emotion recognition accuracy on the INTERFACE for the Castilian Spanish database. In comparison, emotion recognition on MESD reached 93.3% and 89.4% for male and female voices, respectively. In addition, accuracy and F-scores were above 70% for all emotions and voices. To that extent and based on supervised learning analysis, the MESD can be considered as a reliable source for Mexican speech emotional stimuli. In addition, the MESD includes child voices for which the SVM validation method yielded 83.3% emotion recognition accuracy.

**Table 10.** Emotion recognition in speech performances from previous works.

| Reference | Database | Supervised Machine-Learning Approach | Average Emotion Recognition Accuracy |
|---|---|---|---|
| [34] | Berlin Database of Emotional Speech | SVM based on a Gaussian radial basis function kernel with harmonic frequency indices (Fourier Parameters) as inputs | 88.9% |
| | CASIA Chinese emotional corpus | | 79% |
| | Chinese elderly emotional speech database | | 76% |
| [35] | Berlin Database of Emotional Speech | SVM on modulation spectral and frequency features, energy cepstral coefficients, frequency weighted energy cepstral coefficients, and MFCC based on reconstructed signal | 86.2% |
| | INTERFACE for Castilian Spanish | | 90.4% |
| Present work | MESD | Cubic SVM with spectral, prosodic and voice quality features as inputs | 88.9% |
| | INTERFACE for Castilian Spanish | | 90.2% |

High emotion recognition performance has been reached by recording four times more utterances and voices than the number of observations finally integrated into the database, so that only utterances that best discriminated single emotions were selected. This methodology guarantees that recordings resulting from lower actor performance were not considered, and simulation biases were faded. Furthermore, a highly representative Mexican way of expressing emotions is ensured by selecting actors according to daily-life and educational cultural environments. In that sense, the goal of this study was to guarantee the emotional discrimination across the MESD. That is why we implemented the SVM-based validation method, and the high performance for emotion recognition on the final version of the MESD is the reflection of this process. The emotion recognition performance on utterances from each actor was presented, but the utterances finally integrated into the MESD were selected to guarantee the emotional discrimination from future listeners.

It is important to note that the creation of the MESD relied on the predictive model (i.e., cubic SVM) used to select the most representative emotional utterances that integrated the database. Therefore, if using another supervised learning algorithm, other utterances would have been inserted into the MESD. Nevertheless, the intrinsic properties of the SVM to handle small datasets and performance on the final version on the MESD confirmed the reliability of the SVM-based validation method. Furthermore, similar results have been reached on the INTERFACE for Castilian Spanish database which confirmed the reliability of this supervised learning model to handle this type of dataset.

Moreover, clustering analysis performed on child voice recordings led to low silhouette values when considering all the actors' utterances. The weak cluster structure highlights a slight tendency toward two ways of uttering emotions across actors, although all utterances were representative of the Mexican cultural shaping of emotional prosodies. During childhood, the development of emotional intelligence (i.e., the ability to understand, monitor, discriminate and manage emotions) is shaped by parenting styles that depend on parents' levels of demandingness and responsiveness [36]. For instance, parents who prefer communication, acceptance of child reasoning and justification of parental expectation (authoritative parenting) foster social and emotional skills contrary to those who practice coercive control and adult authority without reasoning (authoritarian parenting) [36]. Music and physical education may also shape emotional skills. Indeed, music and speech share common acoustic patterns that underlie emotional communication [37], converting music training into a significant factor of emotional awareness and expressivity development [38]. Furthermore, the practice of physical and sport activities may involve coping with personal

improvement, failure and success, self-regulation, empathy and communication with peers, favoring emotional understanding expression, and management skills [39]. Although education styles are correlated to cultural values [40], slight differences might appear from one household to another. Consequently, in the present study, those factors may have led to slight within-culture differences across child actors in the way they expressed emotions.

### 3.2. Emotion Induction

To date, supplies for emotional speech stimuli adapted to Mexican Spanish are very scarce. Caballero-Morales et al. [9] created a simulated emotional speech corpus for anger, happiness, neutral and sadness that contains 10 sentences per emotion uttered by three males and three females from Huajapan de Leon in Oaxaca, Mexico. Sentences varied from one emotion to another so that their content was congruent with the emotion conveyed. Emotion recognition by supervised learning performed with Hidden Markov Models with MFCC, energy, delta and acceleration coefficients as inputs reached more than 87% accuracy for each emotion. On the other hand, the EmoWisconsin database, elaborated by Pérez Espinosa and colleagues [41], contains emotional utterances from 28 Mexican children of both sexes aged between 7 and 13 recorded in a non-isolated room. Emotions were induced by playing a card-sorting game with an adult examiner who created emotional environments, so that the child vocally expressed positive and negative emotions. Utterances were retrospectively classified into "undetermined", "doubtful", "annoyed", "motivated", "nervous", "neutral" and "confident" and evaluated according to valence and arousal dimensions. However, the EmoWisconsin database presented weak inter-evaluator agreement and low classification, as well as regression performances for emotional evaluation. More recently, [42] elaborated the Interactive Emotional Children's Speech Corpus which is composed of emotional speech segments uttered by 174 male and female children aged between 6 and 11. Emotions were induced by interactions with a robot during a game controlled in a Wizard-of-Oz setting that was designed to induce positive and negative emotions, prospectively categorized into anger, fear, happiness, certainty, sadness, thoughtfulness, surprise and disgust. This database has the particularity of being the largest emotional speech-corpus of robot–child interaction for Mexican Spanish and offers a wide variety of voices. Nevertheless, the inter-evaluator agreement on emotion categorization was low (half of utterances resulted in discrepancies among evaluators when considering the entire dataset), which lowered its reliability.

One crucial difference between previous speech emotional databases for Mexican Spanish and the MESD relies on the nature of the emotion induction. Sentences may be preferred because of their higher ecological validity compared to single words. However, the emotion may not be stressed in each word with the same degree, introducing variations in emotional information throughout the sentence [43]. Most of all, listeners' accuracy, confidence levels, and reliance on particular acoustic cues during emotion vocal identification may differ between sentences and words. A recent study highlighted that for shorter stimuli such as words, temporal cues (duration) may best predict the level of listeners' confidence in emotion recognition, whereas spectral cues (pitch, jitter) may best predict confidence ratings for emotional sentences [44]. Furthermore, the emotional comprehension through words and sentences relies on different time-dynamic processes. Sentence comprehension involves prediction, integration of words and syntactic unification processes that may interplay with the processing of the emotional information [45,46]. Therefore, word stimuli may be adequate to evaluate the emotional processing of the listener without cofounding factors related to the integration of lexical-syntactic information.

The communication of emotional information by prosody is generally accompanied by congruent emotional semantics. The listener is therefore required to integrate information from both channels (prosody and semantics) to understand the emotional utterance [47]. Previous works focused on the time course of neuronal integration of both emotional channels during speech comprehension suggested that the semantic information cannot be ignored even if not voluntarily attended and is coupled with the prosodic information

approximately 400 ms after the onset of the stimulus [47,48]. Incongruencies between semantics and prosody lead to higher processing load and difficulties to understand the emotional content [48]. As the cognitive integration processes governing both emotional prosody alone and emotional prosody with semantics are underlined by different mechanisms, the use of pseudo-words or pseudo-sentences (semantically meaningless artificial language) may be preferred if trade-off effects between semantics and prosody are required to be vanished [44,48]. Nevertheless, pseudo-language is less ecologically valid as it generally does not appear in real-life conversations. In the MESD, two corpora for single-word emotional utterances are proposed: one composed of words controlled for emotional semantics, so that the incongruency effect on the listener's emotional understanding is faded (corpus B), and another one composed of words that recur across emotions (corpus A).

### 3.3. Effect of Controlling Familiarity, Frequency, and Concreteness

From statistical comparisons between the emotional utterances of words coming from corpus A and from corpus B, a significant influence of controlling words' familiarity, frequency of use, and concreteness was observed on the spectral, prosodic and voice quality features of emotional utterances for both adult (male and female) and child voices. Previous studies have highlighted the effect of those linguistic parameters on emotion perception.

The concreteness (as opposed to abstractness) of the concept denoted by a word is part of the semantic information [49]. Abstract concepts, whose understanding is shaped by the internal affective experience, provide greater emotional associations than words that identify concrete objects or actions [50]. Abstractness and emotional information yield the enhanced allocation of processing resources both at early perceptual and late stages of word comprehension [49–51]. The level of concreteness may therefore be a cofounding factor in the listener's understanding of the emotional utterance. Furthermore, the access to emotional information may also be interfered by the lexical representation of the word [52]. The higher salience denoted by more frequent words may interact with the cognitive resources allocation required to process emotional stimuli [53]. As a matter of fact, interactions between word frequency and emotion have been denoted at early and late stages of the processing of emotional information. Scott and colleagues observed enhanced electroencephalographical neuronal activity between 135 and 180 ms (N1 component) after the onset of low-frequency neutral words as compared to emotional words [52]. On the contrary, for high-frequency words, negative stimuli induced higher neuronal activity than neutral and positive words. In [53], low-frequency negative words elicited higher neuronal activity than low-frequency neutral nouns approximately 450 ms after the stimulus onset (P450), whereas this was not observed for high-frequency words. To evaluate the listener's emotional comprehension, it is therefore crucial to consider emotional speech as linguistic information and to control trade-off effects between linguistic and emotional processing. What is more, familiarity is an index of the subjective individual experience with the word [54]. Previous studies have highlighted the existence of a correlation between the familiarity and the emotional perception of the stimulus: familiar words tend to be perceived as more positive than unfamiliar ones [54,55]. Within the present work, those findings are complemented by stressing that familiarity, frequency of use, and concreteness not only modulate emotional perception, but also its vocal expression. The MESD proposes a word corpus controlled for those parameters, which should facilitate the design of future investigations on listeners' cognitive processing of affective prosody.

### 3.4. Cultural Variations in Emotional Prosodies

The universality hypothesis argues that across cultures, perceivers should be able to recognize emotions with accuracies far above chance, which is in favor of a universal way of conveying and understanding emotions [56,57]. To test this statement, Laukka and colleagues [58] used machine learning computation to classify emotions based on speech segments uttered by professional actors from five different English-speaking cultures (Australia, India, Kenya, Singapore, and the United States (US). They trained an SVM with

a radial basis kernel function on 30 features that denote temporal, spectral and prosodic acoustic information. The classifier was tested on information extracted from either the same or a different culture than the one used in the training phase. In line with the universality hypothesis, emotion recognition was above chance level in all conditions, including when cultures were not matched between training and testing phases. This result provides evidence for fundamental acoustic cues shared across cultures to express emotions [59]. However, accuracy was higher when the classifier was trained and tested on data extracted from speech segments uttered by speakers from the same culture. This observation emphasizes a cultural shaping of affective prosody [58,60]. For instance, Singaporean people tend to express anger with a higher pitch and more intensity than English-speakers from Kenya [5]. In the present work, statistical comparisons between Castilian and Mexican Spanish emphasized consistent differences regarding the spectral, prosodic and voice quality features of emotional utterances.

Owing to the lack of Castilian Spanish emotional utterances for child voices, cultural shaping during childhood has not been tested. However, throughout development, socialization experiences transmit emotional practices relevant to cultural values [40]. A recent study explored child and mother behaviors during 10 min of free-play interactions and used the Early Childhood Behavior Questionnaire for parents to assess their 2-year-old child temperament [61]. They compared emotional and social interactions shaped by US and German cultures and observed more enthusiastic and higher emotional tones in US mother–child exchanges than German ones. Parents from the two cultures varied in their expectation for particular emotional expressivity during socialization, with German parents being satisfied with less enjoyment during play [61]. These miscellaneous socialization goals have shaped the child's emotional development and reinforce cultural gaps for emotional expressivity and understanding since early childhood. Neonates (12–72 h old) have demonstrated that they respond to emotional speech uttered in their mother native language but did not when emotional speech was uttered in a different language. This suggests that cultural learning of emotional expressivity in speech may even start during the fetal period [62]. Although further investigations are still needed, the cultural shaping of emotional prosody during all developmental stages is expected to take place from childhood to adulthood.

Chronaki and colleagues explored cross-cultural emotion recognition in voice by children (8.5–10.5 years old), adolescents (11–13 years old) and adults (19–35 years old) [63]. Participants had to recognize emotional prosodies (happiness, fear, sadness and neutral) in pseudo-utterances uttered by Canadian (English), Argentine (Spanish), Mandarin (Chinese) and Jordanian/Syrian (Arabic) native speakers. They observed that emotion recognition improved with age in their native language (English), but not systematically in non-native ones, with greater progress from adolescence to adulthood than from childhood to adolescence. From these observations, the authors suggested that later developmental stages during adolescence (after the age of 13) are crucial in the development of emotion recognition in voice abilities. Moreover, the lack of improvement from childhood to adulthood for non-native cultures highlighted specialization within a single ethnic group to recognize culturally shaped socio-emotional rules. The increase in social exploration behaviors fostered by the neuroplastic maturation of "social brain" networks during adolescence [64] may underlie better accuracy among older adolescents and adults to recognize affective prosodies. Lower classifier performances for child voices than adult ones reported in the present study are in line with those developmental trajectories. Prior literature on emotion recognition was complemented by outlining the improvement of emotional expressivity in speech from childhood to adulthood.

### 3.5. Value of the Data and Contributions

- The presented dataset provides a Mexican cultural shaping of emotional linguistic utterances with adult male, adult female and child voices for six emotional states: (1) anger, (2) disgust, (3) fear, (4) happiness, (5) neutral and (6) sadness. The MESD

seems to be the first set of single-word emotional utterances that includes both adult and child voices for the Mexican population.
- Engineers and researchers can use this database to train predictive model algorithms for emotion recognition. For instance, it could help smart healthcare systems for classification of emotional speech-related diseases such as depression or autism [65], or to help identify specific personality traits from speech [66].
- The MESD may be used as auditive and linguistic emotional stimuli for the exploration of emotional processing in healthy and/or pathological populations.
- The MESD provides emotional utterances from two corpora: (corpus A) nouns and adjectives that are repeated across emotional prosodies and types of voice (female, male and child), and (corpus B) words controlled for age-of-acquisition, frequency of use, familiarity, concreteness, valence, arousal and discrete emotion dimensionality ratings.
- Results from statistical analysis confirmed the existence of trade-off effects between words' emotional semantics, frequency, familiarity, concreteness, and emotional prosodies expression, as well as prosodic cultural variations between Mexican and Castilian Spanish.

## 4. Materials and Methods

This study is the first stage of a project registered at the Biomedical Center under the following number: ISRCTN18117434 [67] and is part of the realization of the study protocol published under the following https://doi.org/10.3389/fnhum.2021.626146 (accessed on 15 November 2021) [68].

### 4.1. Acquisition of the Castilian Spanish INTERFACE Database

The INTERFACE database is composed of 100 affirmative sentences, 34 interrogative and stressed sentences, 16 paragraphs, 10 digits and 24 isolated single words. For our work, single-word utterances in which 24 words recurred across six emotional prosodies were considered. These were: (1) anger, (2) disgust, (3) fear, (4) joy, (5) neutral/normal, and (6) sadness. The INTERFACE database includes voices from one male and one female adult speaker. Speech contents (i.e., words and emotional prosodies) do not differ from one gender to the other. Furthermore, two identical recording sessions per speaker are available. In sum, 288 single-word utterances per gender are included (2 times the repetition of the 24 words across the 6 emotional prosodies).

The Castilian Spanish database for emotional and neutral utterances was acquired by academic request to the Evaluation and Language resources Distribution Agency.

### 4.2. MESD Speech Corpus

The written corpus was obtained by selecting words from two sources: (1) the single-word corpus from the INTERFACE for Castilian Spanish database, named corpus A, and (2) the Madrid Affective Database for Spanish (MADS) [54,69]. The corpus of words selected from MADS was named corpus B. For the creation of the MESD, the 24 isolated words from INTERFACE were selected to create corpus A. Those words recurred across voices (child, adult female and male) and emotions (anger, disgust, fear, happiness, neutral and sadness).

The MADS shares ratings for psycholinguistic variables of 875 words rated by both female and male adults. Words for corpus B were selected according to their grammatical class and ratings for valence, arousal, discrete emotional dimension (anger, disgust, fear, happiness and sadness), concreteness, familiarity, frequency of use and subjective age of acquisition. In particular, all words were nouns or adjectives, with subjective age of acquisition strictly under 9. Words for neutral utterances were selected to have valence and arousal ratings strictly greater than 4 but strictly lower than 6 (on a 9-point scale), whereas words for other emotional utterances had valence and arousal ratings ranging from 1 to 4, or from 6 to 9. A rating higher than 2.5 (on a 5-point scale) for a particular emotion

allowed the qualitative categorization of the word for the corresponding anger, disgust, fear, happiness or sadness prosody.

Words for corpus B were selected so that emotions were matched for concreteness, familiarity and frequency of use ratings. Word selection was carried out separately for male, female and mean ratings for all subjects. Namely, a one-way ANOVA with emotion as a factor was conducted on each parameter separately (concreteness, familiarity and frequency). Independence of residuals was assessed by the Durbin–Watson test. Normality and homogeneity were assessed by the Shapiro–Wilk test and Bartlett tests, respectively. In case of non-parametricity, a Kruskal–Wallis test was applied. Post hoc tests were used to statistically assess specific differences (Tukey after ANOVA, Wilcoxon tests with *p*-value adjustment by the Holm method after Kruskal–Wallis). Level of significance was set at $p < 0.05$. In case of significance, outliers were removed until non-significance was reached. All ratings for frequency, familiarity or concreteness that were outside the range defined by percentiles 2.5 and 97.5 were considered as outliers. Statistical analysis was conducted using R software (R Foundation for Statistical Computing, Vienna, Austria).

In sum, the total speech corpus was composed of 48 words per emotion (24 from corpus A and 24 from corpus B) so that 288 single words were included for further utterance by male, female or child voices. Words in Spanish along with their English translation are detailed in Appendix A. Tables A1 and A2 present words of corpus A and B, respectively.

### 4.3. Participants for MESD Setting-Up

Four male adults (mean age = 22.75, SD = 2.06), four female adults (mean age = 22.25, SD = 2.50), and eight children (five girls and three boys, mean age = 9.87, SD = 1.12) voluntarily participated in emotional speech recordings. Participants were included if they had grown up in Mexico in a cultural Mexican environment (Mexican academic education and family environments). Participants were excluded if they presented any pathology that affects emotional behavior, hearing, speech, or sickness traits affecting voice timbre. No participant had lived in a foreign country (other than Mexico) for more than two weeks in the last four years.

### 4.4. Material and Procedure for Recording MESD

Recordings were carried out in a professional recording studio with the following materials: (1) a Sennheiser e835 microphone with a flat frequency response (100 Hz to 10 kHz), (2) a Focusrite Scarlett 2i4 audio interface connected to the microphone with an XLR cable and to the computer, (3) audio files generated were recorded in the digital audio workstation REAPER (Rapid Environment for Audio Production, Engineering, and Recording). They were stored as a 24-bitsequence with a sample rate of 48,000 Hz.

Adult and child sessions lasted one hour, and 30 min, respectively, and took place following a one-on-one format. Adults were asked to utter 288 words (48 words per emotion), whereas children had to utter 144 words (24 words per emotion). Four children uttered the words from corpus A, and four children uttered the ones from corpus B. The order of corpora was counterbalanced across adult sessions. Namely, two male adults and two female adults were asked to firstly utter corpus A and then corpus B. Next, two male adults and two female adults were asked to utter firstly corpus B, and then corpus A. Emotions were randomly distributed for both cases: adults and children.

Participants were given as much time as they required to read and get a grip on the entire word dataset before uttering each word with the intended emotional intonations: neutral, happy, angry, sad, disgusted or afraid. Participants were asked to do their best to get into the emotional role. All words from a particular emotion were uttered successively to facilitate this process. Participants were asked to wait at least five seconds between two utterances in order to focus before each recording. See Figure 9 for a graphical representation of the procedure followed on individual sessions with each actor.

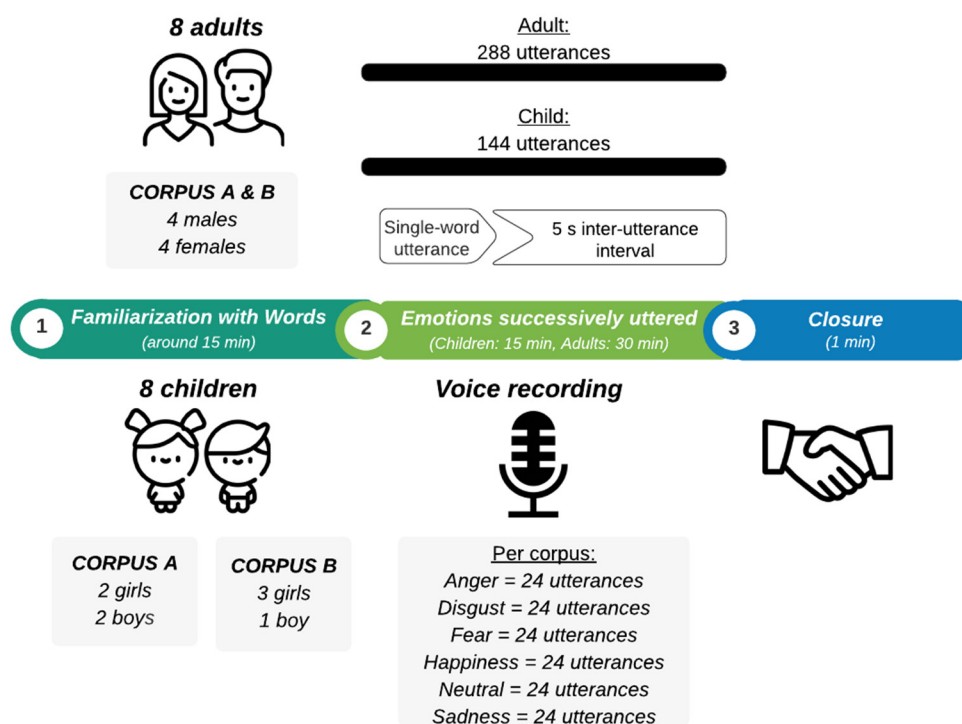

**Figure 9.** Procedure for one-on-one recording sessions. Adult actors uttered 288 words (i.e., 24 words per emotion from each corpus) and children uttered 144 words (i.e., 24 words per emotion from corpus A or B). For adult sessions, the order of emotion was random, and the order of corpora was counterbalanced across speakers. For child session, the order of emotion was random across actors.

### *4.5. MESD Validation Based on Supervised Identification of Emotional Patterns*

To evaluate the affective prosody of recorded voices, a traditional supervised machine-learning algorithm was proposed. In this way, emotion detection was quantified (at least in terms of machine-learning logistics), and classification performance could be a measurement of the expected emotional detection a posteriori. In addition to the MESD evaluation, the same analysis was applied on the INTERFACE voice recordings in order to compare the level of affective prosody in terms of the proposed evaluation method (only for adult voices since INTERFACE does not provide child voices). This process validation proceeds as described below.

#### 4.5.1. Acoustic Feature Extraction

Before extracting acoustic features from MESD utterances, each word was excerpted from the continuous recording of each session in order to generate an audio file for each individual word. The acoustic features of interest for pursuing emotion recognition are detailed in Table 11. This information was extracted individually for each word using Praat and Matlab R2019b.

#### 4.5.2. Data Normalization

The resulting 30 acoustic features were normalized to reduce inter-individual biases due to physical anatomy (e.g., body corpulence). For this purpose, acoustic features were rescaled between 0 and 1 according to the max–min normalization as described in Equation (1). That is,

$$x_{normalized} = \frac{x - min_k}{max_k - min_k} \tag{1}$$

where $x$ is the feature to be normalized, $max_k$ is the highest value of acoustic feature vector $k$ and $min_k$ is the lowest value of such vector.

**Table 11.** Extracted acoustic features for emotion recognition.

| Type | Feature | Description | Mathematical Expression |
|---|---|---|---|
| Prosodic | Fundamental frequency or pitch (Hertz) | Mean and standard deviation over the entire waveform. | $\mu_1 = \sum_{n=0}^{N-1} x(n)/N$ <br> $\sigma_1 = \sqrt{\sum_{n=0}^{N-1}(x(n)-\mu)^2/N}$ <br> where $\mu_1$ is the mean, $\sigma_1$ is the standard deviation, $N$ is the total number of samples and $n$ is a sample of signal $x$. |
| | Speech rate | Number of syllables per second. | |
| | Root mean square energy (Volt) | Index of the energy of the signal over the entire waveform. | $\sqrt{\sum_{n=0}^{N-1}|x(n)|^2/N}$ <br> where $N$ is the total number of samples and $n$ is a sample of signal $x$. |
| | Intensity (dB) | Mean and standard deviation over the entire waveform. | $\mu_2 = \sum_{n=0}^{N-1} X(n)/N$ <br> $\sigma_2 = \sqrt{\sum_{n=0}^{N-1}(X(n)-\mu)^2/N}$ <br> where $\mu_2$ is the mean, $\sigma_2$ is the standard deviation, $N$ is the total number of samples and $X(n)$ is the intensity in dB at sample $n$. |
| Voice quality | Jitter (%) | Jitter local: average absolute difference between two consecutive periods, divided by the average period. <br> Jitter ppq5: 5-point period perturbation quotient. It is the average absolute difference between a period and the average of it and its four closest neighbors, divided by the average period. | $\left(\frac{\sum_{i=2}^{P}|T_i - T_{i-1}|}{(P-1)} \Big/ \frac{\sum_{i=1}^{P} T_i}{P}\right) \times 100$ <br> where $T_i$ is the duration of $i$th period, and $P$ is the number of periods. <br><br> $\left(\frac{\frac{\sum_{i=3}^{P-2}|T_i - (T_{i-2}+T_{i-1}+T_i+T_{i+1}+T_{i+2})/5|}{(P-4)}/P-4}{\frac{\sum_{i=1}^{P} T_i}{P}}\right) \times 100$ <br> where $T_i$ is the duration of $i$th period, and $P$ is the number of periods. |
| | Shimmer (%) | Shimmer local: the average absolute difference between the amplitude of two consecutive periods, divided by the average amplitude. <br> Shimmer rapq5: 5-point amplitude perturbation quotient. It is the average absolute difference between the amplitude of a period and the average of the amplitude of it and its four closest neighbors, divided by the average amplitude. | $\left(\frac{\sum_{i=2}^{P}|A_i - A_{i+1}|}{(P-1)} \Big/ \frac{\sum_{i=1}^{P} A_i}{P}\right) \times 100$ <br> where $A_i$ is the duration of $i$th period, and $P$ is the number of periods. <br><br> $\left(\frac{\frac{\sum_{i=3}^{P-2}|A_i - (A_{i-2}+A_{i-1}+A_i+A_{i+1}+A_{i+2})/5|}{(P-4)}/P-4}{\frac{\sum_{i=1}^{P} A_i}{P}}\right) \times 100$ <br> where $A_i$ is the duration of $i$th period, and $P$ is the number of periods. |
| | Mean harmonics-to-noise ratio (HNR; dB) | Relation of the energy of harmonics against the energy of noise-like frequencies. | If 99% of the signal is composed of harmonics and 1% is noise, then HNR is defined by: <br> $10\log_{10}(99/1) = 20$ dB <br> Mean harmonics-to-noise ratio between time point $t_1$ and time point $t_2$ is defined by: <br> $\frac{1}{(t_2 - t_1)} \int_{t_1}^{t_2} dt\, x(t)$ <br> where $x(t)$ is the HNR (in dB) as a function of time. |
| Spectral | Formants (Hertz) | F1, F2, F3: Mean and bandwidth in center. | |
| | MFCC | 1–13 coefficients. | Conversion to Mel scale following: $f_{Mel} = 2595\log_{10}\left(\frac{f_{linear}}{700} + 1\right)$ <br> where $f_{Mel}$ is the frequency in Mel scales, and $f_{linear}$ is the frequency in the linear scale (Hertz). |

### 4.5.3. SVM-Based Validation Method

Matlab R2019b was used to carry out a supervised machine-learning analysis based on SVM predictive models. For this work, a classification algorithm that can be effective in high-dimensional spaces on relatively small sample sizes and with a low sensitivity to outlier values was needed. SVM particularly fits this description [70]. This method is based on establishing a threshold in a dimensional space to separate classes accurately. The distance between the threshold and the observations is called the classification margin. SVM has a low sensitivity to the training data; namely, outlier values are misclassified, which increases the model bias during training. In this case, the margin is called the soft classification margin and all observations at the edge or within the soft margin are called support vectors. The advantage of this model is that by allowing misclassifications during training, the system has a higher performance during validation on unknown data (lower variance). When the dataset has more than three dimensions, the Support Vector Classifier is a hyperplane. SVM handles overlapping classifications by the computation of the relationship between real values and their corresponding values if they were mapped to another dimension that is defined by the kernel function. For instance, the cubic SVM calculates the relationship between the real values and their cubic transformation, so that they can be accurately classified in flat affine subspaces [71].

Hyperparameters were adjusted to a cubic kernel function and a box constraint level (soft-margin penalty) adjusted to 10. The multiclass method (one-vs-one or one-vs-all) and the kernel scale parameters was set to "auto", meaning that the algorithm was automatically optimized for both parameters according to the dataset. The dataset was divided into two groups: training and validation, corresponding to 77% and 23% of the data, respectively. A stratified train/test split holdout cross-validation method was used, so that each group presented an equal number of words per emotion. Six parameters were computed in order to evaluate the classifier performance: (1) accuracy, which is the ratio between number of tuples correctly classified and the total number of tuples; (2) precision, which represents the relation of the number of correctly classified positive tuples against the total number of tuples classified as positive, including true and false positives (it is an index of exactness of the predictive model); (3) false discovery rate, which is the ratio between the number of incorrectly classified negative tuples and the total number of tuples classified as positive; (4) recall, which represents the relation of the number of positive tuples correctly classified against the total number of positive tuples, including true positives and false negatives (this is an index of the completeness of the predictive model); (5) false negative rate, which is the ratio between the number of positive tuples incorrectly classified, and the total number of positive tuples; and (6) F-score, which is the harmonic mean of precision and recall (it reflects the balance between both parameters). This machine-learning procedure was the core of the validation process. Algorithmic adjustments were undertaken for adult and child voices. Such adjustments are specified below.

### 4.5.4. Machine Learning Procedure for Adult Voices

For adult voices, the process of validation included the comparison between MESD and INTERFACE in terms of classification performance on the basis of the proposed SVM-based validation method. For both databases (MESD and INTERFACE), the emotion labelling during training was based on the emotion intended to be expressed during recordings. Female and male voices were analyzed independently. The input data for training was the 30 normalized acoustic features extracted from each utterance after a dimensionality reduction based on Principal Component Analysis (PCA), explaining 95% of the variance [20]. An individual classification process was conducted for each actor (INTERFACE: 1 male, 1 female; MESD: 4 four males, 4 females), that is, 288 utterances (48 for each emotion). The final version of the MESD was created by selecting for each emotion the utterances from the actor leading to the highest F-score during validation. Figure 10 details this process. The resulting set of 288 utterances was used to evaluate the accuracy and F-score for emotion recognition on the final version of MESD.

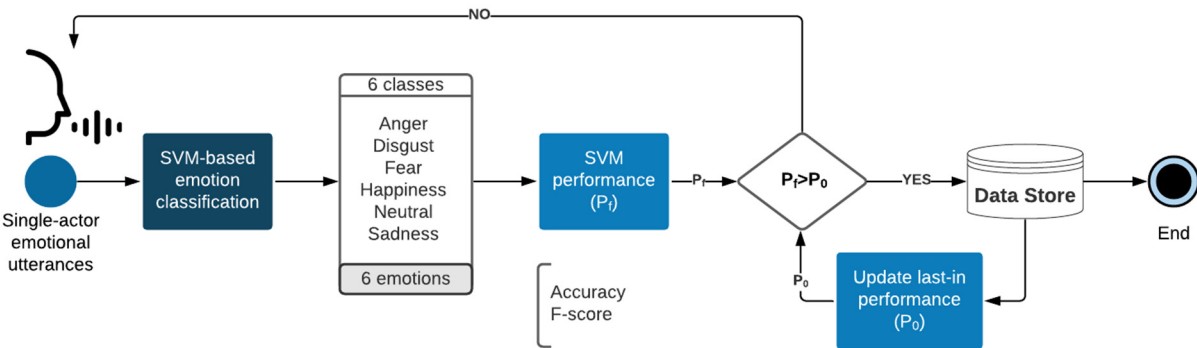

**Figure 10.** Selection process for adult voices from MESD. For the process, utterances from each participant were classified per emotion. The utterances of the participants who reached the highest F-score for each emotion were kept in the MESD.

### 4.5.5. Machine-Learning Procedure for Child Voices

For child voices, the process validation did not include the comparison between MESD and INTERFACE since the latter database did not have available child recording voices.

The machine-learning procedure for child voices in MESD proceeded as follows. Firstly, a k-mean clustering analysis was applied on the 30 normalized acoustic features extracted from utterances of each emotion separately (24 observations per participant, leading to 6 datasets of 192 observations). Squared Euclidean distance metrics and k-means + + algorithms for cluster-center initialization were used. The optimized number of clusters was assessed by computing silhouette scores. The number of clusters that led to the highest average silhouette score was selected; namely, 2 clusters. Figure 11 details silhouette values from 2 to 8 clusters on each emotion.

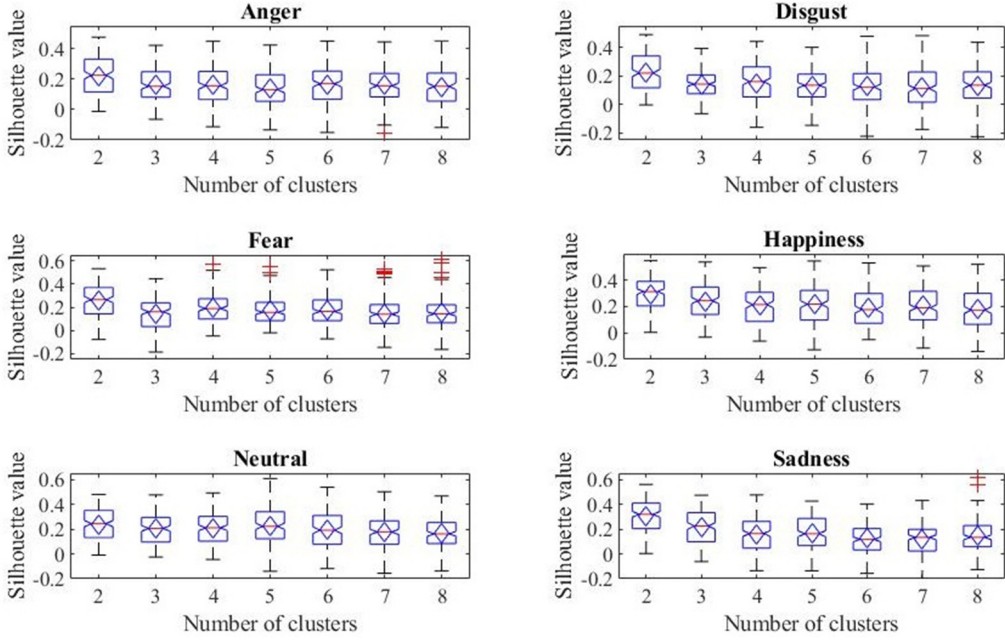

**Figure 11.** Silhouette values from two to eight clusters on each emotion. Diamonds are average values. Red central lines indicate the median. Bottom and top edges of the box correspond to the 25th and 75th percentiles, respectively. Outliers are plotted individually using the red '+' symbol.

In each cluster, utterances of words coming from corpus A (4 participants) and corpus B (4 participants) were considered separately. For utterances from both corpora, the number of observations for individual participants in each cluster was computed. Pairs of participants (one who uttered words from corpus A and one who uttered words from corpus B) were assessed in each cluster by considering the participant with the highest number of

observations. As a result, each pair of participants was composed of 288 utterances (48 per emotion, including 24 of words from corpus A and 24 of words from corpus B).

Then, a classification process was conducted on the basis of the SVM-based validation method. The input data for training was the 30 normalized acoustic features extracted from each utterance, after a dimensionality reduction based on PCA that explained 95% of the variance. Emotion classification was carried out on data from each resulting pair. The final version of the MESD was created by selecting for each emotion the utterances from the pair leading to the highest F-score during validation. Figure 12 details this process. The resulting set of 288 utterances was used to evaluate the accuracy and F-score for emotion recognition on the final version of MESD.

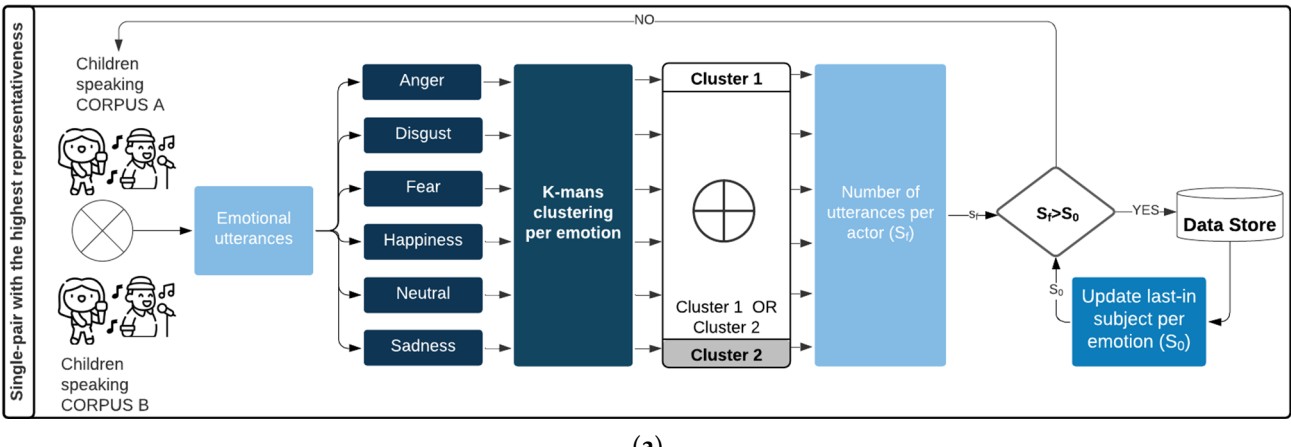

(**a**)

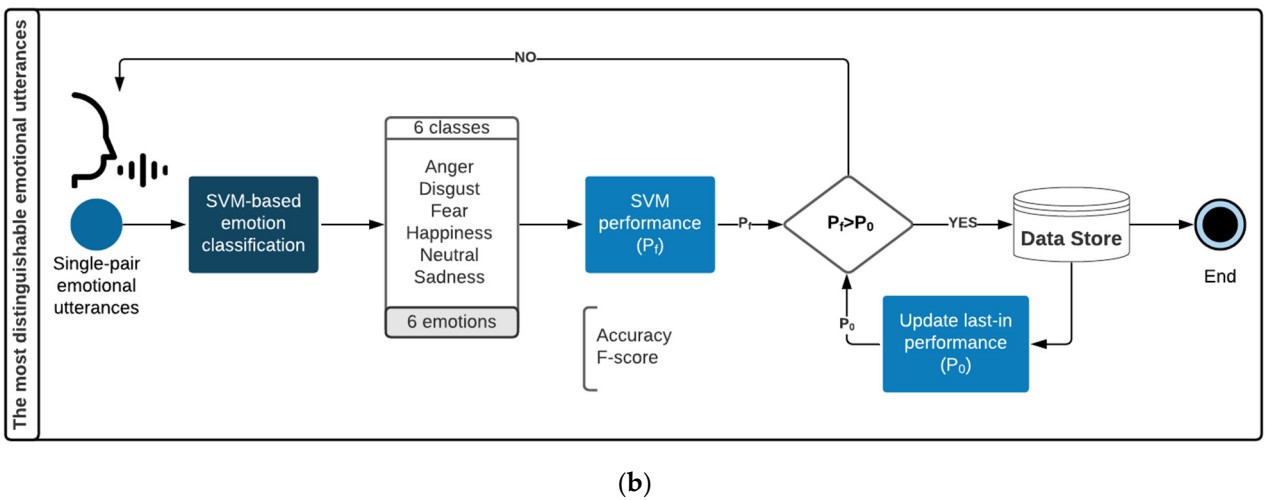

(**b**)

**Figure 12.** Selection process for child voices. For the process, utterances from individual actors were clustered per emotion (**a**). Utterances from each most representative pair of participants (corpus A/corpus B) were classified per emotion. The utterances of the pair of participants who reached the highest F-score for each emotion were kept in the MESD (**b**).

*4.6. Statistical Analysis*

In addition to the SVM-based validation method, a statistical analysis was performed with R software (R Foundation for Statistical Computing, Vienna, Austria). Level of significance was set at $p < 0.05$, regardless of the applied method. Before the statistical analysis, parametricity was tested, and then, a parametric or non-parametric method was selected. For all statistical analysis, data related to Mexican-shaped emotional utterances came from the final version of MESD established after the emotion classification, specified previously in Figures 10 and 12.

4.6.1. Inter-Emotion and Inter-Culture Comparisons

The aim was to emphasize the acoustic signature of emotion discrimination in both Mexican and Castilian cultures separately. Male, female and child voices were analyzed independently. Statistical tests were conducted on normalized acoustic features separately. Data were normalized according to Equation (1), and acoustic features were those mentioned in Table 11, except for MFCC. For emotions comparisons, 288 observations were considered (48 per emotion) in each database. Normality and homogeneity were assessed using Shapiro–Wilk and Bartlett tests, respectively. Owing to the non-parametric distribution of the data, a Kruskal–Wallis test was applied. Post hoc comparisons were conducted by Wilcoxon tests with *p*-value adjustment by the Holm method.

Thereafter, the aim was to highlight the cultural shaping of emotional prosodies. For culture comparisons, only the data from corpus A were considered to allow comparison with the Castilian Spanish dataset. Therefore, 144 observations were considered (24 per emotion) in each database. The observations for the INTERFACE database were extracted randomly from session 1 or 2 to complete the same word corpus as MESD. Similarly, statistical tests were conducted on the normalized acoustic features independently. Owing to the non-parametrical distribution of data and inefficacy of data transformation, Friedman tests with the interaction between emotion and culture as a factor were conducted for each extracted acoustic feature and voice (male and female) separately. For post hoc comparisons, Conover test with *p*-value adjustment by the Holm method was applied.

4.6.2. Effects of Word Familiarity, Frequency, and Concreteness

We aimed to assess the effect of controlling frequency, familiarity, and concreteness on affective expressions. Namely, we compared utterances from MESD of words from corpus A versus words from corpus B. Statistical tests were conducted on the normalized acoustic features independently. Male, female, and child voices were considered separately. Owing to the non-parametrical distribution of data and inefficacy of data transformation, Kruskal–Wallis tests were conducted with interaction between emotion and culture as factors for each extracted acoustic feature independently. Post hoc comparisons were conducted by Wilcoxon tests with *p*-value adjustment by Holm method.

**5. Conclusions**

The Mexican Emotional Speech Database (MESD) seems to be the first public available database to propose a Mexican cultural shaping of adult and child voices for emotional utterances of single-words. The MESD contains affective expressions for anger, disgust, fear, happiness, neutral and sadness. Within the framework of previous works for Mexican Spanish, the present study adds the following contributions: (1) emotional utterances from both adults (male and female) and children voices were recorded; and (2) the MESD proposes a word corpus controlled for emotional meaning, frequency of use, familiarity and concreteness, as well as another one composed of words recurring across voices and emotions. Results from classification analysis underlined competitive performances that validate the MESD reliability compared with other databases. From statistical analysis, significant cofounding effects of words emotional meaning, frequency, familiarity and concreteness were outlined on spectral, prosodic and voice quality features in emotional utterances. Finally, cultural variations in emotional prosodies were explored by comparing emotional utterances from Mexican and Spanish native speakers. In line with the dialect theory, a cultural shaping of emotional expressivity in speech was highlighted. The MESD complements the current data repository of Mexican-shaped affective utterances and is a reliable source to be used for further investigation, health care, or acoustic engineering purposes.

**Author Contributions:** Conceptualization, M.M.D.; methodology, M.M.D., L.M.A.-V. and D.I.I.-Z.; software, M.M.D.; validation, M.M.D., L.M.A.-V. and D.I.I.-Z.; formal analysis, M.M.D.; investigation, M.M.D.; resources, M.M.D.; data curation, M.M.D.; writing—original draft preparation, M.M.D.; writing—review and editing, M.M.D., L.M.A.-V. and D.I.I.-Z.; visualization, M.M.D.; supervision, L.M.A.-V. and D.I.I.-Z.; project administration, M.M.D., L.M.A.-V. and D.I.I.-Z.; funding acquisition, M.M.D. All authors have read and agreed to the published version of the manuscript.

**Funding:** This work was supported by Mathilde's (M.M.D.) PhD scholarship sponsored by the Mexican National Council of Science and Technology (reference number: 1061809). The sponsor was not involved in the conduct of the research and/or preparation of the article, nor in the decision to submit the article for publication.

**Institutional Review Board Statement:** The study was conducted according to the guidelines of the Declaration of Helsinki and approved on 14 July 2020, by the Ethics Committee of the School of Medicine of Tecnologico de Monterrey (register number within the National Committee of Bioethics CONBIOETICA 19 CEI 011-2016-10-17) under the following number: P000409-autismoEEG2020-CEIC-CR002.

**Informed Consent Statement:** A written informed consent was obtained from all participants. In the case of children, their parents authorized their participation.

**Data Availability Statement:** The data presented in this study are openly available in Mendeley Data at http://doi.org/10.17632/cy34mh68j9.1 (last access: 15 November 2021) [72].

**Acknowledgments:** All authors are grateful to the "Instituto Estatal de la Juventud (INJUVE)" that allowed us to access the recording studio for free. We would like to thank Norberto E. Naal-Ruiz (https://orcid.org/0000-0002-1203-8925, accessed on 15 November 2021) who lent us the recording equipment. We acknowledge the ELDA (Evaluation and Language resources Distribution Agency) S.A.S. for sharing the "Emotional speech synthesis database, ELRA catalogue (http://catalog.elra.info, accessed on 15 November 2021), ISLRN: 477-238-467-792-9, ELRA ID: ELRA-S0329".

**Conflicts of Interest:** The authors declare no conflict of interest.

## Abbreviations

| | |
|---|---|
| HNR | Harmonics-to-noise ratio |
| MESD | Mexican Emotional Speech Database |
| MFCC | Mel-Frequency Cepstral Coefficients |
| PCA | Principal Component Analysis |
| SVM | Support Vector Machines |

## Appendix A

**Table A1.** Word corpus A for emotional utterances. English translations are provided.

| Corpus A | | | | | |
|---|---|---|---|---|---|
| **Emotion** | **Type of Voice for Emotional Utterance** | **Spanish** | | **English Translation** | |
| Anger | | Abajo | Basta ya | Below | Enough |
| | | Ayer | Arriba | Yesterday | Above |
| Disgust | | Fuera | Gracias | Outside | Thank you |
| | | Hola | Por favor | Hello | Please |
| Fear | Adult female | Hoy | De nada | Today | Your welcome |
| | Adult male | No | Izquierda | No | Left |
| Happiness | Child | Sí | Derecha | Yes | Right |
| | | Adiós | Dentro | Goodbye | Within |
| Neutral | | Nunca | Pronto | Never | Soon |
| | | Lento | Rápido | Slowly | Quick |
| Sadness | | Tarde | Detrás | Afternoon/Late | Behind |
| | | Antes | Delante | Before | Ahead |

**Table A2.** Word corpus B for emotional utterances. English translations are provided.

| Corpus B | | | | | |
|---|---|---|---|---|---|
| **Anger** | **Disgust** | **Fear** | **Happiness** | **Neutral** | **Sadness** |
| Adult Female | | | | | |
| Relajación | Relajación | Tristeza | Calmado | Agencia | Tristeza |
| Amenazado | Abuso | Abuso | Cómico | Barba | Relajación |
| Insultado | Oruga | Delincuencia | Delfín | Caducado | Abandono |
| Atasco | Creído | Amenazado | Estudioso | Religioso | Ceguera |
| Ceguera | Monstruoso | Explosivo | Fresco | Consciente | Injusticia |
| Odioso | Cucaracha | Araña | Bebé | Pelirrojo | Delincuencia |
| Torturado | Delincuencia | Furia | Clavel | Contrario | Abuso |
| Conflicto | Amenazador | Ceguera | Masaje | Desierto | Monstruoso |
| Abuso | Horroroso | Relajación | Oasis | Extenso | Desmayo |
| Delincuencia | Injusticia | Desmayo | Refrescante | Acuático | Enfriamiento |
| Explosión | Atasco | Horroroso | Delicadeza | Bola | Explosivo |
| Amenazador | Insultado | Dificultad | Calma | Fecha | Fracasado |
| Monstruoso | Jeringuilla | Engañado | Rosa | Acuarela | Hambre |
| Explosivo | Monstruo | Amenazador | Siesta | Garganta | Cansancio |
| Furia | Araña | Estricto | Ternura | Temporal | Odioso |
| Huracán | Conflicto | Aguijón | Chimenea | Giro | Huracán |
| Ataque | Náusea | Explosión | Velas | Hondo | Insultado |
| Estricto | Oloroso | Cucaracha | Afición | Mentiroso | Llanto |
| Injusticia | Puñalada | Fantasma | Aglomeración | Elevado | Metralleta |
| Ira | Engañado | Alarma | Fantástico | Metro | Explosión |
| Irrespetuoso | Robo | Huracán | Atractivo | Mojado | Mortal |
| Engañado | Abandono | Cirugía | Aventurero | Paella | Náusea |
| Ofensa | Chistoso | Conflicto | Baile | Mecánico | Conflicto |
| Pesadilla | Torturado | Infarto | Bello | Presumido | Ofensa |
| English translation | | | | | |
| Relaxation | Relaxation | Sadness | Calmed | Agency | Sadness |
| Threated | Abuse | Abuse | Comedian | Beard | Relaxation |
| Insulted | Caterpillar | Crime | Dolphin | Expired | Abandonment |
| Traffic jam | Vain | Threated | Studious | Religious | Blindness |
| Blindness | Monstrous | Explosive | Fresh | Aware | Injustice |
| Hateful | Cockroach | Spider | Baby | Redhead | Crime |
| Tortured | Crime | Fury | Carnation | Opposite | Abuse |
| Conflict | Threatening | Blindness | Massage | Desert | Monstrous |
| Abuse | Dreadful | Relaxation | Oasis | Wide | Faint |
| Crime | Injustice | Faint | Refreshing | Aquatic | Cooling |
| Explosion | Traffic jam | Dreadful | Fineness | Ball | Explosive |
| Threatening | Insulted | Difficulty | Calm | Date | Loser |
| Monstrous | Syringe | Deceived | Rose | Watercolor | Hunger |
| Explosive | Monster | Threatening | Nap | Throat | Tiredness |
| Fury | Spider | Strict | Tenderness | Temporary | Hateful |
| Hurricane | Conflict | Sting | Fireplace | Turn | Hurricane |
| Attack | Nausea | Explosion | Candles | Deep | Insulted |
| Strict | Odorous | Cockroach | Hobby | Liar | Cry |
| Injustice | Stab | Ghost | Conglomerate | High | Machine gun |
| Rage | Deceived | Alarm | Fantastic | Subway | Explosion |
| Disrespectful | Theft | Hurricane | Attractive | Wet | Mortal |
| Deceived | Abandonment | Surgery | Adventurous | Paella | Nausea |
| Offense | Humorous | Conflict | Dance | Mechanic | Conflict |
| Nightmare | Tortured | Heart attack | Beautiful | Boastful | Offense |
| Adult Male | | | | | |
| Aguijón | Aguijón | Inconsciente | Antiguo | Alto | Funeraria |
| Farsa | Amenazado | Amenazador | Ducha | Artículo | Tristeza |
| Ansiedad | Celda | Delincuencia | Baile | Ferrocarril | Abandono |
| Conflicto | Monstruo | Ataque | Aventurero | Átomo | Celda |

**Table A2.** *Cont.*

| Corpus B | | | | | |
|---|---|---|---|---|---|
| **Anger** | **Disgust** | **Fear** | **Happiness** | **Neutral** | **Sadness** |
| Delincuencia | Operación | Celda | Pelirrojo | Extenso | Danza |
| Ataque | Desorden | Araña | Respiración | Babosa | Dificultad |
| Dificultad | Caprichoso | Danza | Siesta | Chulo | Antiguo |
| Encadenado | Estafa | Aguijón | Tranquilidad | Consciente | Ira |
| Estafa | Araña | Enojo | Confidente | Delgado | Encadenado |
| Abandono | Metralleta | Cirugía | Agua | Desierto | Amenazado |
| Furia | Abandono | Estafa | Satisfacción | Abierto | Furia |
| Hambre | Cucaracha | Furia | Broma | Estación | Delincuencia |
| Desorden | Mutilado | Dificultad | Afición | Despistado | Estafa |
| Huracán | Náusea | Conflicto | Calmado | Arroz | Ansiedad |
| Danza | Delincuencia | Ansiedad | Ensueño | Escalera | Hambre |
| Amenazador | Odioso | Huracán | Calma | Azulejo | Huracán |
| Impaciencia | Eructo | Imprudente | Alegría | Colorado | Infarto |
| Enojo | Puñalada | Abandono | Fresco | Estrecho | Conflicto |
| Disputa | Sangre | Cucaracha | Amistad | Agencia | Enojo |
| Amenazado | Terrorismo | Encadenado | Espléndido | Botón | Injusticia |
| Injusticia | Encadenado | Infarto | Atractivo | Congelado | Llanto |
| Atasco | Furia | Amenazado | Relajación | Escurridizo | Locura |
| Ira | Monstruoso | Explosión | Astuto | Fecha | Ataque |
| Malévolo | Torturado | Ira | Bebé | Bola | Metralleta |
| English translation | | | | | |
| Sting | Sting | Unconscious | Antique | Tall | Funeral parlor |
| Sham | Threated | Threatening | Shower | Article | Sadness |
| Anxiety | Cell | Crime | Dance | Railway | Abandonment |
| Conflict | Monster | Attack | Adventurous | Atom | Cell |
| Crime | Operation | Cell | Redhead | Wide | Dance |
| Attack | Mess | Spider | Breathing | Slug | Difficulty |
| Difficulty | Capricious | Dance | Nap | Pimp | Antique |
| Chained | Scam | Sting | Tranquility | Aware | Rage |
| Scam | Spider | Annoyance | Informer | Thin | Chained |
| Abandonment | Machine gun | Surgery | Water | Desert | Threatened |
| Fury | Abandonment | Scam | Satisfaction | Open | Fury |
| Hunger | Cockroach | Fury | Joke | Station | Crime |
| Mess | Amputee | Difficulty | Hobby | Absentminded | Scam |
| Hurricane | Nausea | Conflict | Calmed | Rice | Anxiety |
| Dance | Crime | Anxiety | Daydream | Stairs | Hunger |
| Threatening | Hateful | Hurricane | Calm | Tile | Hurricane |
| Impatience | Burp | Imprudent | Happiness | Red-colored | Heart attack |
| Annoyance | Stab | Abandonment | Fresh | Narrow | Conflict |
| Argument | Blood | Cockroach | Friendship | Agency | Annoyance |
| Threated | Terrorism | Chained | Magnificent | Button | Injustice |
| Injustice | Chained | Heart attack | Attractive | Frozen | Cry |
| Traffic jam | Fury | Threatened | Relaxation | Slippery | Madness |
| Rage | Monstrous | Explosion | Clever | Date | Attack |
| Wicked | Tortured | Rage | Baby | Ball | Machine gun |
| Child | | | | | |
| Amenazado | Abuso | Desmayo | Acogedora | Botón | Cansancio |
| Catástrofe | Repulsivo | Abuso | Clavel | Congelado | Desmayo |
| Explosivo | Monstruoso | Amenazador | Afición | Delgado | Tristeza |
| Estricto | Atasco | Catástrofe | Fresco | Espacial | Traición |
| Ceguera | Crisis | Ceguera | Masaje | Extenso | Ataque |
| Injusticia | Delincuencia | Delincuencia | Oasis | Bola | Catástrofe |
| Abandono | Estafa | Araña | Baile | Ferrocarril | Humillación |
| Robo | Aguijón | Cirugía | Relax | Hondo | Ceguera |
| Dañino | Furia | Alarma | Fantástico | Labrador | Injusticia |
| Ataque | Humillación | Conflicto | Celebración | Mecánico | Dañino |

**Table A2.** *Cont.*

| Corpus B | | | | | |
|---|---|---|---|---|---|
| **Anger** | **Disgust** | **Fear** | **Happiness** | **Neutral** | **Sadness** |
| Delincuencia | Injusticia | Humillación | Chistoso | Elevado | Abandono |
| Abuso | Abandono | Abandono | Atractivo | Pelirrojo | Disputa |
| Amenazador | Suspenso | Disputa | Bebé | Colorado | Enojo |
| Disputa | Insultado | Estricto | Coordinación | Rizado | Amenazado |
| Enojo | Monstruo | Injusticia | Bello | Rueda | Explosión |
| Condena | Mutilado | Robo | Calmado | Seco | Abuso |
| Estafa | Cucaracha | Aguijón | Aventurero | Escurridizo | Estafa |
| Alterado | Araña | Alterado | Rosa | Temporal | Condena |
| Explosión | Náusea | Cucaracha | Cita | Vapor | Explosivo |
| Dificultad | Amenazador | Dañino | Aplausos | Mojado | Furia |
| Furia | Odioso | Estafa | Creativo | Estrecho | Robo |
| Impaciencia | Irrespetuoso | Condena | Danza | Húmedo | Conflicto |
| Atasco | Puñalada | Explosión | Delicioso | Robusto | Delincuencia |
| Humillación | Robo | Amenazado | Carcajada | Mimado | Infarto |
| English translation | | | | | |
| Threatened | Abuse | Faint | Cozy | Button | Tiredness |
| Catastrophe | Repulsive | Abuse | Carnation | Frozen | Faint |
| Explosive | Monstrous | Threatening | Hobby | Thin | Sadness |
| Strict | Traffic jam | Catastrophe | Fresh | Spatial | Treason |
| Blindness | Crisis | Blindness | Massage | Wide | Attack |
| Injustice | Crime | Crime | Oasis | Ball | Catastrophe |
| Abandonment | Scam | Spider | Dance | Railway | Humiliation |
| Theft | Sting | Surgery | Relax | Deep | Blindness |
| Harmful | Fury | Alarm | Fantastic | Farmer | Injustice |
| Attack | Humiliation | Conflict | Celebration | Mechanic | Harmful |
| Crime | Injustice | Humiliation | Humorous | High | Abandonment |
| Abuse | Abandonment | Abandonment | Attractive | Redhead | Argument |
| Threatening | Fail | Argument | Baby | Red-colored | Annoyance |
| Argument | Insulted | Strict | Coordination | Curly | Threatened |
| Annoyance | Monster | Injustice | Beautiful | Wheel | Explosion |
| Condemnation | Amputee | Theft | Calmed | Dry | Abuse |
| Scam | Cockroach | Sting | Adventurous | Slippery | Scam |
| Annoyed | Spider | Annoyed | Rose | Temporary | Condemnation |
| Explosion | Nausea | Cockroach | Date | Steam | Explosive |
| Difficulty | Threatening | Harmful | Applause | Wet | Fury |
| Fury | Hateful | Scam | Creative | Narrow | Theft |
| Impatience | Disrespectful | Condemnation | Dance | Moist | Conflict |
| Traffic jam | Stab | Explosion | Delicious | Robust | Crime |
| Humiliation | Theft | Threatened | Guffaw | Spoiled | Heart attack |

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
