# Peer review of "Mexican Emotional Speech Database Based on Semantic, Frequency, Familiarity, Concreteness, and Cultural Shaping of Affective Prosody"

_data, 2021_

Round 1

Reviewer 1 Report

This paper proposes a new voice database which is named Mexican Emotional Speech Database (MESD). The database contains various emotions such as anger, disgust, fear, happiness, neutral, and sadness, with adult (male and female) and child voices. The paper is well organized and well structured. The paper can be improved based on the following comments:

1) In the paper, more recent papers should be cited, particularly from 2021, because there is only one cited paper from 2021.

2)  The results section can be improved by comparing the SVM with the other advanced classifiers such as Random Forest, ANN, CNN etc.

Reviewer 2 Report

Major Problem with the paper:

-- Experiments should be conducted speaker-independent approach. If experiments are speaker-independent, the accuracies will come down. The present results are very high and they are not trustable. 

-- Results based on individual dataset, gender etc., are missing. Also, include confusion matrices. 

My other concerns are: 

1) English is somewhat poor and needs to checked by professional english reader. 

2) clarity of figures is missing. 

3) Related work is not coherent, just listed few papers. Needs to be discussed coherently and may be you can use several references to write paragraphs. 

4) Missing lot of references which exploits features based on speech production, such as "Analysis of Emotional Speech - A Review", Toward Robotic Socially Believable Behaving Systems - Volume I : Modeling Emotions, Springer International Publishing, pp. 205 - 238, March 2016.

Authors should cite all the references shown above and also recent publications published in INTERSPEECH, ICASSP and IEEE Trans. on Affective computing. 

5) Section 2: Details about the databases is missing such as number of speakers, geneder-wise, duration of utterances, distribution of the data according to emotion. Also, listing the details in a tabular/histogram form is better. For example, see: Naturalistic Audio-Visual Emotion Database. Proc. 11th ICON, 1, 175-182, 2014. 

6) Need to highlight Contributions and which should be clear and compact. List them in bullet form. 

7) Abbreviations can be listed as table. 

8) Description about mel scale may not be necessary and you should compress/remove some such things throughput the paper. 

9) How the experimentation is carried out is not clear. Make use of tables or illustrations. 

10) It appears experiments are not conducted in speaker-independent approach. If conducted, the accuracies will come down. The present results are very high and they are not trustable. 

11) Describe clearly all the captions to the tables and figures. So that, by reading them itself readers should understand the essence. Also, use consistent format for denoting (a), (b) ...etc.

Round 2

Reviewer 1 Report

The authors improved the quality of the paper and addressed my comments in the paper extensively. The paper is now ready for publication.

Reviewer 2 Report

All the comments raised in my previous review have been carefully considered and addressed by the authors.